# Combining BGC-Argo floats and satellite observations for water column estimations of particulate backscattering coefficient

Jorge García-Jiménez[1], Ana B. Ruescas[1], Julia Amorós-López[1], and Raphaëlle Sauzède[2]

[1]Image Processing Laboratory, Universitat de València, Spain
[2]Institut de la Mer de Villefranche, FR3761, CNRS, Sorbonne Université, France

**Correspondence:** Jorge García-Jiménez (jorge.garcia-jimenez@uv.es)

**Abstract.** As the second largest carbon reservoir on Earth, the ocean regulates the carbon balance through dissolved and particulate organic carbon forms. Monitoring carbon cycle processes is key to understanding the climate system. Although most organic carbon in the ocean exists in dissolved form, Particulate Organic Carbon (POC), despite its smaller share, plays a vital role by connecting surface biomass production with the deep ocean and sedimentation processes. POC estimation is achieved by measuring proxies like the Particulate Backscattering Coefficient ($b_{bp}$) estimated from satellite observations and *in situ* sensors, such as the BioGeoChemical-Argo (BGC-Argo) floats. Previous studies have integrated data from BGC-Argo floats and satellite sensors, demonstrating the potential of machine learning models to estimate vertical bio-optical properties within the water column. The approach presented here enhances the estimation within the top 250 meters of the water column compared with previous works. The estimations are performed in two distinct regions-the North Atlantic and the Subtropical Gyres; and across several layers within two maximum depth limits, 50 m and 250 m. Data from BGC-Argo profiles and the Ocean and Land Colour Instrument (OLCI) sensor are used together to build a training dataset for the Random Forest model, which is applied with different sets of variables. Additional considerations regarding our datasets include short time criteria for match-ups ($\pm 24h$) and high spatial resolution. The Random Forest model shows promising results, especially within the first 50 meters in the Subtropical Gyres.

## 1 Introduction

The ocean covers approximately 70% of Earth's surface and plays a fundamental role in regulating climate dynamics. It redistributes energy and carbon through a variety of physical and biogeochemical processes. Among these processes, the biological carbon pump facilitates the transfer of $CO_2$ from the atmosphere to the ocean floor by enabling the production and sinking of particulate organic carbon (POC), which is sequestered in deep-ocean sediments. POC originates from living organic carbon, primarily produced by photosynthetic organisms such as phytoplankton, which thrive in the sunlit upper ocean layers. These organisms require carbon compounds, along with light and nutrients, to survive and reproduce (Falkowski et al., 1998; Siegel et al., 2014). Their presence and abundance reflects the interplay of resources and losses in the environment (Behrenfeld et al., 2006), with populations maintaining daily division cycles even in regions where nutrients appear to be depleted beyond detection limits (Ribalet et al., 2015; Vaulot and Marie, 1999). Quantifying phytoplankton biomass and carbon content

is crucial to understanding these ecosystem dynamics and their role in carbon cycling. Chlorophyll-a (chl-a) concentration has traditionally served as a proxy for phytoplankton biomass, but its interpretation is often challenged by physiological photo-acclimation, which alters intracellular pigment levels without necessarily reflecting actual changes in biomass. The Particulate Backscattering Coefficient ($b_{bp}$) has been recognized as a stable optical proxy for phytoplankton biomass and carbon content as it is sensitive to the abundance, size distribution, and composition of suspended particles, rather than pigment concentration alone (Behrenfeld and Boss, 2006; Graff et al., 2015; Martinez-Vicente et al., 2013). Unlike chl-*a*, which can underestimate biomass in stratified and oligotrophic waters, $b_{bp}$ remains relatively unaffected by photo-acclimation effects, making it particularly useful for studying carbon fluxes across different oceanic regions and depth layers. The complex interaction between key variables (usually non-linear) and limited sampling resolution in dynamic environments, combined with the technical challenges of depth-resolved measurements, contribute to gaps in our understanding of specific marine processes, such as carbon sequestration, nutrient cycling, sedimentation and the ocean-atmosphere $CO_2$ exchange.

Bio-optical sensors installed on autonomous platforms, such as the Biogeochemical-Argo (BGC-Argo) profiling floats (Claustre et al., 2020), have become a valuable technology for acquiring *in situ* data about the water masses ecological and physical status. These sensors can measure the scattering of light in water, which provides information about radiative transfer conditions and the nature and dynamics of suspended particulate matter. The $b_{bp}$ parameter is an inherent optical property (IOP) of water, and it has been widely recognized as a robust bio-optical proxy for POC (Cetinić et al., 2012; Sullivan et al., 2013). However, $b_{bp}$ measured by floats can have an uncertainty of the order of 10–15% (Bisson et al., 2019). These uncertainties stem from the instrumental drift, the sensor calibration limitations, and the reliance on manufacturer calibration files rather than sensor-specific calibrations using dark counts. While autonomous platforms provide extensive spatial and temporal coverage, these factors must be considered when interpreting bio-optical data to ensure accuracy and reliability.

IOPs are intrinsic characteristics of water, determined solely by its composition and are independent of the external light field or the geometrical angle conditions during observation. These properties include absorption, elastic scattering, inelastic processes (such as fluorescence and Raman scattering), and attenuation, which describe how light behaves and propagates through water. IOPs are essential in studying light interactions in aquatic environments, as they reflect the presence of dissolved organic matter, phytoplankton and suspended particles. The $b_{bp}$ can be measured by autonomous platforms spread out across the ocean; or derived by scatter measurements from onboard satellite sensors, such as the Sentinel-3 Ocean and Land Colour Instrument (OLCI)[1] (EUMETSAT, 2019; Jorge et al., 2021; Koestner et al., 2024). Designing observational strategies based on combining the two approaches constitutes a fundamental tool for improving knowledge of ocean processes (BGC, 2016).

Several approaches have been developed to estimate POC from optical measurements of water leaving radiance ($L_w$), or by linking POC with remote sensing derived IOPs (Bisson et al., 2019; Evers-King et al., 2017; Loisel et al., 2002; Stramski et al., 2008). However, these methods are designed to estimate parameters at the sea surface, which does not fully capture the complexities of carbon export in the ocean, as numerous vertical processes within the water column significantly influence the carbon cycle. Fusing satellite data with vertical profiles from BGC-Argo floats to extend the measurements of surface bio-optical properties (i.e. $b_{bp}$) to several depth layers is performed with the SOCA method in Sauzède et al. (2016, 2020). The

---

[1] https://sentinel.esa.int/web/sentinel/user-guides/sentinel-3-olci/product-types/level-2-water

initial SOCA2016 method consists of a neural network combining satellite surface estimates of $b_{bp}$ and chl-$a$ concentration,
matched up in space and time with depth-resolved physical properties derived from temperature-salinity profiles measured by
BGC-Argo profiling floats. This method predicts $b_{bp}$ for 10 different depths in the productive layer. In 2020, the availability of a
larger database with new profiles and the opportunity to increase the vertical resolution of model outputs led to the development
of the SOCA2020 method. This approach includes additional Sea Level Anomaly (SLA) inputs with information about sub-
mesoscale processes; it replaces satellite-derived products ($b_{bp}$ and chl-$a$) by simple reflectances at several wavelengths and
explores machine learning-based techniques that are efficient at estimating retrievals, in addition to quantifying the uncertainty
associated with the outputs. A significant improvement in the $b_{bp}$ predictions was revealed, especially near the surface layers.

Building on these results, this research proposes an analysis of the $b_{bp}$ estimation in the upper layers of the ocean surface
using the Sentinel-3 Ocean and Land Colour Instrument (S3OLCI). We change the spatial resolution from the 4 km resolution
of GlobColour level-3 merged products (1/24° at the equator) used in previous studies, with the 300 m Full Resolution (FR) of
Sentinel-3 OLCI. Additionally, we evaluate the model performance after incorporating OLCI spectral wavelengths as features
for $b_{bp}$ estimation and compare these results with those obtained using GlobColour. Another key aspect of this study is deter-
mining whether adding IOPs derived from satellite data (absorption and scattering) improves the accuracy of the $b_{bp}$ estimation
compared to using reflectances alone. Furthermore, $b_{bp}$ at different depths of the water column is estimated using multi-output
models. These multi-output random forest models account for the high correlation between measurements at nearby depths.
Finally, there is a comparison of the accuracy of the $b_{bp}$ estimations in two depth limits, that is from the surface to either 50 m
or 250 m.

## 2  Data and methods

Data from *in situ* measurements collected by BGC-Argo floats, along with satellite data from various projects and missions
(GlobColour and Sentinel-3 OLCI) are utilized as inputs for the machine learning models. We employ three datasets for two
different maximum depths—50 m and 250 m. The three datasets are: 1) Level-3 multi-sensor products from GlobColour; 2)
Level-2 single-sensor reflectances from Sentinel-3 OLCI processed with the Case 2 Regional Coast Colour (C2RCC) algorithm
(Brockmann et al., 2016); and 3) The second dataset plus derived IOPs from OLCI using again the C2RCC processor.

### 2.1  Study Area

Two regions of the ocean are analyzed, the North Atlantic (NA), within latitudes 35°–80°N, and the Subtropical Gyres (STG),
within latitudes 15°–40° North and South (see Figure 1). These two areas exhibit distinct seasonal patterns throughout the year,
experiencing significant differences in terms of nutrients, light availability, minimum and maximum temperature regimes,
Mixed Layer Depth (MLD) variations, thermocline levels, and mesoscale dynamics. One of the main differences between
these two regions is the variability in the stratification of the upper ocean layers. This phenomenon determines the resistance
of the water to overturning, thus conditioning the supply of nutrients from deeper waters (Lozier et al., 2011). NA waters are
seasonally high in chl-$a$ (mg.m$^{-3}$). During winter, a weakly stratified upper ocean water column overturns or mixes, facilitating

the upwelling of nutrients needed to sustain surface productivity. In the STG region, spanning thousands of kilometers across the oceans, nutrients are in short supply, and waters range from ultra-oligotrophic (chl-a $\leq$ 0.04 mg.m$^{-3}$) to oligotrophic (chl-a $\leq$ 0.07 mg.m$^{-3}$) (Letelier et al., 2004). During the summer and winter cycles, there is expansion and contraction of their spatial coverage, respectively (Leonelli et al., 2022). Despite these extreme nutrient limitations, molecular clock studies have shown that phytoplankton in these regions continue to divide daily, suggesting that microbial communities have adapted through efficient nutrient recycling, regenerated production, and physiological acclimation strategies (Vaulot, 1995; Ribalet et al., 2015). Feucher et al. (2019) showed that the two Northern Hemisphere subtropical gyres have qualitatively very similar stratification structures, with permanent pycnoclines in the North Atlantic and North Pacific.

These regional differences in physical and biological characteristics are also reflected in the vertical distribution of the $b_{bp}$, with NA exhibiting higher surface variability and deeper gradients compared to the more stable stratification of the STG (see Figure 1). The NA, with its seasonal mixing and higher productivity, generally exhibits higher $b_{bp}$ values due to increased particulate matter and phytoplankton-derived organic material in the water column. In contrast, the STG, characterized by strong stratification and lower phytoplankton biomass, show significantly lower $b_{bp}$ values, indicative of reduced particle concentrations. Despite the global coverage of the STG sampled regions, there is much more heterogeneity in the NA observations, making it a more complex and challenging environment for modeling purposes, as observed in the results.

The temporal distribution of the match-ups shows a clear seasonal bias, with most data concentrated between May and September, particularly during 2017. This uneven distribution is primarily due to the limited availability of cloud-free satellite observations required to match with BGC-Argo profiles, especially during winter months when cloud cover and low solar angles reduce the quality of remote sensing products.

## 2.2 BGC-Argo Data

The international One-Argo program provides continuous ocean observations through an array of profiling floats, each equipped with sensors tailored to specific objectives: Core-Argo (for temperature and salinity measurements); BGC-Argo (for biogeochemical measurements); Deep-Argo (for measurements deeper than 2,000 m); and Polar-Argo (for measurements in polar environments). Key bio-optical variables, such as chlorophyll-a, optical particulate backscattering, and irradiance, can be measured using BGC-Argo profiling floats. These variables are essential for generating products that support biogeochemical and ecosystem studies (Claustre et al., 2009, 2020). The BGC-Argo floats can collect measurements from 1,000 m to the surface, with a depth resolution of $\sim$1 meter, every 10 days, even though in many cases the vertical resolution is poorer.

The lower boundary of the euphotic zone is defined as the depth where 1% of the Photosynthetically Available Radiation (PAR) penetrates the water column (Kirk, 1976). While it is true that phytoplankton growth is ultimately driven by absolute photon flux rather than a relative threshold (Sverdrup, 1953; Behrenfeld and Boss, 2017), the 1% PAR definition remains a widely used metric for characterizing the physical light environment across diverse oceanographic conditions. This definition focuses on describing the light field as a physical property rather than directly linking it to biological responses, providing a consistent, measurable boundary for analyzing water column dynamics. While recent studies have demonstrated that phytoplankton can grow at light levels significantly below this threshold or even in polar night conditions (Randelhoff et al., 2020),

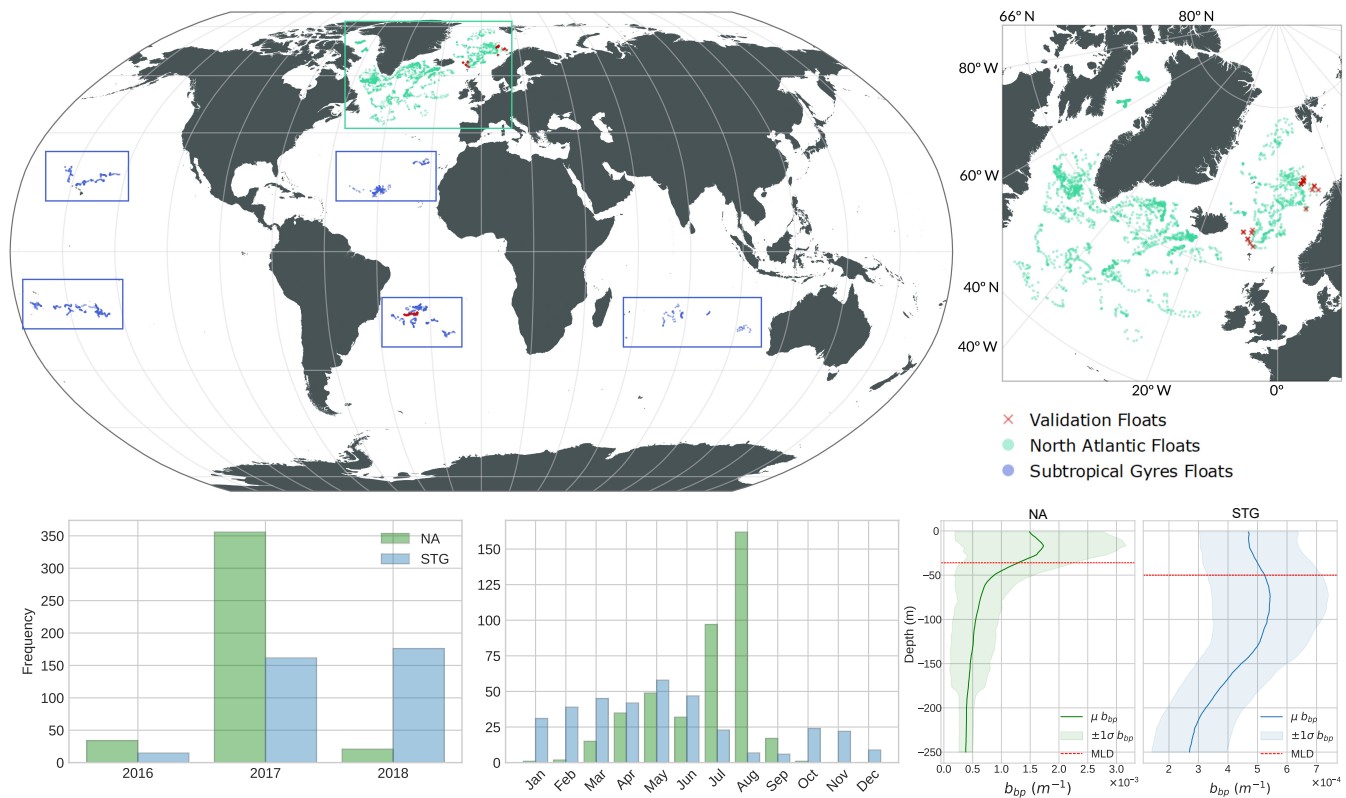

**Figure 1.** Global map showing the geographic locations of the BGC-Argo floats and satellite data match-ups. [Botttom row] Temporal coverage of match-ups by year (left) and month (middle) for the North Atlantic (NA, green) and Subtropical Gyres (STG, blue). Vertical profiles (right) of $b_{bp}$ from floats, where solid lines show mean values, shaded areas $\pm 1$ standard deviation, and the dashed red line the average Mixed Layer Depth (MLD).

our physical optics approach allows for standardized comparisons of light attenuation patterns across our datasets. This depth varies in the global ocean from $\sim$20 m to more than 120 m, depending on the region and season. The flux of sinking carbon that exits the euphotic zone due to gravity is a key component of the overall carbon sequestration budget (Siegel et al., 2014). In the present experiments, a depth limit extending beyond the lower boundary of the euphotic zone (250 m depth) was selected. From 250 meters to the surface, measurements of temperature, salinity, density, and spiciness were taken every 2 meters, along with information about the MLD -calculated as the depth at which density exceeds 0.03 kg m$^{-3}$ relative to the density at 10 m (de Boyer Montégut et al., 2004). Vertical measurements of $b_{bp}$ at the same vertical resolution are also available in the datasets. Spiciness reflects density-compensated variations in temperature and salinity, providing a tracer for water mass origins and mixing processes (Smith and Ferrari, 2009). Since particle concentrations and optical properties often differ between water masses, spiciness anomalies can be associated with variations in the $b_{bp}$. Warmer and saltier waters (higher spiciness) can stratification, reducing vertical nutrient fluxes and potentially limiting biological production, leading to lower concentrations

of organic particulate matter and thus lower the $b_{bp}$. Table 1 shows the different types of variables used to train and validate the proposed models for the designed experiments.

The $b_{bp}$ value (Mignot et al., 2014) used here is calculated following the work of Sullivan and Twardowski (2009). The angular distribution of scattering relative to the direction of light propagation $\theta$ at the optical wavelength $\lambda$ is known as the
140 volume scattering function (VSF), $\beta(\theta, \lambda)$ (m$^{-1}$, sr$^{-1}$). It is composed of the sum of pure sea water $\beta_{sw}$ and particles $\beta_p$, where $\beta_{sw}$ depends on temperature and salinity, and is calculated using a depolarization ratio of 0.039 (Zhang et al., 2009). The contribution of $\beta_p$ to the VSF is calculated subtracting the contribution of $\beta_{sw}$ from $\beta(124°, \lambda)$:

$$\beta_p(124°, \lambda) = \beta(124°, \lambda) - \beta_{sw}(124°, \lambda) \tag{1}$$

Then, a conversion factor $\chi$ with a value 1.076 for an angle of 124° relates $b_{bp}$ to $\beta_p$, making it possible to extrapolate the
145 measurement from a single angle (124°) to the total coefficient as follows (Boss and Pegau, 2001; Sullivan and Twardowski, 2009):

$$b_{bp}(\lambda) = 2\pi\chi(\beta(\theta, \lambda) - \beta_{sw}(\theta, \lambda)) \tag{2}$$

The backscattering sensor of the BGC-Argo floats measures $\beta(124°, \lambda)$ with $\lambda$ = 700nm. The quality control procedure carried out is the one followed in the SOCA2016 method.

## 2.3 BGC-Argo and Satellite Match-up Databases

The match-up database created for the SOCA2020 experiments, which links BGC-Argo floats with GlobColour and GlobalOcean data, was utilized in this study. The GlobColour data consists of normalized water-leaving reflectances (*rho_wn*) at 5 wavelengths (412, 443, 490, 555 and 670 nm), as well as the Photosynthetically Active Radiation (PAR) product. This *rho_wn* are derived from a combination of sensors that constitute the GlobColour product: SeaWiFS, MERIS, MODIS Aqua,
VIIRS NPP and OLCI (ACRI-ST, 2020). The GlobalOcean set provides Sea Level Anomaly (SLA) data, calculated relative to a 20-year mean of sea surface height, generated with altimeter data from various missions (HY-2A, Saral/Altika, Cryosat-2, Jason-2, Jason-1 T/P, ENVISAT, GFO and ERS1/2) (CMEMS, 2022). In the cited work, the match-up with BGC-Argo floats was performed using the values from the closest available pixels within a $\pm$ 5-day window and on a 5x5 pixel grid. Further details about the procedure can be found in Sauzède et al. (2020).

The BGC-Argo measurements used here were matched with Sentinel-3 OLCI data using the Calvalus tool developed by Brockmann Consult GmbH (Fomferra, 2011). The spatio-temporal approach applied consists of a time window between the BGC-Argo profiles and the satellite measurements of $\pm$ 24 hours, and the spatial satellite coverage around the profile is 3x3 macro pixel on full-resolution imagery (300 m pixel). Once the match-up between satellite and float is established, a baseline quality control is applied to ensure that the satellite-measured reflectances maintain radiometric consistency. First, a flag-based
filter is applied, discarding pixels near or under probable cloudy conditions. This is followed by an outlier removal based on z score ($z = (x - \mu)/\sigma$), applied at macro pixel level band by band. Then, a coefficient of variation in the 560 nm band ($cv = \sigma/\mu$) is applied (Bailey and Werdell, 2006). Coefficient values under 0.2 assure a good spatial homogeneity (Ahmed

**Table 1.** Summary of the variables used in the study

| Data | Description | Variable | Quantity | Variables processed | Type of pre-processing |
|---|---|---|---|---|---|
| **BGC-Argo** | *In situ* sensors | Temperature | 26 (51m)/126 (250m) | 5 | PCA |
| | | Salinity | 26 (51m)/126 (250m) | 5 | PCA |
| | | Density | 26 (51m)/126 (250m) | 5 | PCA |
| | | Spiciness | 26 (51m)/126 (250m) | 5 | PCA |
| | | MLD | 1 | 1 | Standardisation |
| | | Lat/Lon | 2 | 2 | - |
| | | DOY | 1 | 1 | - |
| | | $b_{bp}$ | 26 (51m)/126 (250m) | 26/126 | Stand.+log10 |
| **GlobColour** | Level-3 product | $rho\_wn$ | 5 | 5 | Standardisation |
| | | PAR | 1 | 1 | Standardisation |
| **GlobalOcean** | | SLA | 1 | 1 | Standardisation |
| **Sentinel-3 OLCI** | C2RCC L2 reflectance | $rho\_wn$ | 12 | 12 | Standardisation |
| | C2RCC L2 WQ products | IOPs | 8 | 8 | Standardisation |

et al., 2013; Hlaing et al., 2013; Zibordi et al., 2009). Finally, the median of the pixels left by macro-pixel is used (Hu et al., 2001), which is a standard procedure in studies focused on oceanic waters (Barnes et al., 2019). These criteria reduced the data

set from the original 4115 to 763 match-ups. Specifically, 411 and 352 data points are available for the NA and STG regions. We excluded data from two floats to be used exclusively for validation purposes: in the NA, the float with unique WMO (for World Meteorological Organization Number) 6902545 -with 22 measurements- and in the STG region the float WMO 3902125 -with 28 measurements- constitute the independent dataset in the validation process.

The Sentinel-3 OLCI bands selected extend from 400 nm to 753 nm (bands 1 to 12) of normalized water-leaving reflectances

(*rho_wn*). The extraction is done on level 2 data atmospherically corrected with the Case-2 Regional CoastColour (C2RCC) Processor (Brockmann et al., 2016). C2RCC relies on an extensive database of simulated water-leaving reflectances and related top-of-atmosphere radiances, with neural networks trained to perform inversions both for the atmospheric correction and the in-water quality parameter estimation. C2RCC provides parameters like the absorption and scattering of the different constituents (IOPs) at 443 nm, that is: absorption of chlorophyll pigments (*apig*), yellow substances (*agelb*) and detritus (*adet*); scattering of

particulate matter (*bpart*) and white scatterers (*bwit*), as well as the additive *atot* and *btot*. It also provides total suspended matter concentration, chlorophyll-a concentration, and Apparent Optical Properties (AOPs) like $K_d$ (diffuse attenuation coefficient).

Each parameter has its associated error estimation. From the 25 parameters calculated by C2RCC, we selected the eight IOPs mentioned, plus the reflectance for bands 400 to 753 nm.

## 2.4 Data Pre-processing

The set and number of parameters (measured or derived) available for the experiments is presented in Table 1. The dataset names in the table correspond to the specific features included: GCGO refers to the GlobColour-GlobOcean L3 satellite reflectance, combined with the PAR and SLA products (7 features); BGC denotes the Argo-BGC data after pre-processing (27 features across 26 or 126 layers, depending on the depth of 50 or 250 m, respectively); S3OLCI includes 12 reflectance bands and S3IOPS include the reflectance bands plus the eight C2RCC-derived IOPs. After excluding the measurements for validation, the two areas have a total of 713 inputs. The maximum number of input variables is 37. The size of the matrices can be seen in Table 2. Due to the heterogenous nature of the input variables ($X$) used to train the models and the high dimensionality and covariance of the variables measured along the water column by BGC-Argo floats, the data was preprocessed to reduce redundancy and multicollinearity. The high-dimensional, non-independent variables (temperature, salinity, density, and spiciness) were the ones with the most significant number of features. Each variable had one measurement every 2 meters, which means 126 measurements in the first 250 m, or 26 measurements in the 50 m depth profiles.

To reduce the high dimensionality and simplify the regression models, a Principal Component Analysis (PCA) is applied to some of the input features. After this feature reduction on the high-dimensional variables, the 250 m and 50 m measurements with 126 and 26 inputs are reduced to 5 components for each variable, resulting in a total of 20 features. This method still retains 99% of the information. In addition, satellite-derived variables and the MLD were normalized using zscore standardization, i.e., removing the mean ($\mu_x$) and dividing by the standard deviation ($\sigma_x$) of each feature.

A second preprocessing step consisted of a logarithmic transformation to the $b_{bp}$ values measured by the floats. This compress the dynamic range of the data, typically higher near the surface and decreasing exponentially several orders of magnitude with depth. The transformation reduces the influence of extreme values, particularly near the surface, and helps to stabilize variance across the profiles. As a result, the distribution becomes closer to Gaussian, which facilitates training and improves the robustness of the regression models. Finally, variables that consider the spatio-temporal domain, like latitude, longitude and date (day of year) are also included.

## 2.5 Multi-output machine learning models

There are two main approaches for dealing with multi-output regression problems. One way is to use univariate models, also known as problem transformation methods (Schmid et al., 2022; Borchani et al., 2015). These methods decompose the multi-output regression problem into multiple single-target problems, creating an independent model for each output. The predictions from these separate models are then combined. This approach ignores the relationships between the targets, which can adversely affect the prediction's overall accuracy. Alternatively, multivariate models are designed to capture dependencies and interactions between the outputs, potentially leading to more accurate predictions (Borchani et al., 2015). When and how to apply these two approaches depends on the nature of the data and the correlation between the targets. In our preprocessing

results, PCA decomposition indicates a high covariance among measurements at different depths in the water column. Since our regression models estimate $b_{bp}$ at different depths, it is logical to consider that nearby values in the water column are related to each other.

Random Forest Regressor (RFR) (Breiman, 2001) has been widely applied in geosciences and marine environmental studies for classification and regression tasks (Cutler et al., 2007; Ruescas et al., 2018). Regression trees are at the model's core, which effectively handles complex data when there are non-linear dependencies between a numerical response variable and a diverse set of predictors, whether qualitative or quantitative (D'Ambrosio et al., 2017). RFR is an ensemble method that combines many weak decision tree learners, which are grown in parallel to reduce the bias and variance of the model simultaneously, enhancing the model's predictive performance. Furthermore, RFR provides insights into the importance of the training features, which reveals the variables that have the most significant impact on the predictions. This capability makes the model's mechanisms and results easier to interpret and explain.

Different algorithms have been tested in previous works (see Sauzède et al. (2016, 2020)) to estimate $b_{bp}$ at various depths. Both works are based on a multivariate model applied to all possible outputs. In SOCA16, a Multi-Layer Perceptron is developed, while in SOCA2020 a comparison between a linear model (Ridge) and an ensemble model (Random Forest) is done. The latter showed higher performance. The Multivariate RFR used in this study offers higher accuracy than the univariate RFR, especially when the outputs are highly correlated (Schmid et al., 2022) and when complex interactions demand structured inference to be effectively managed (Xu et al., 2019). All the previously mentioned algorithms, including Linear Regressor (LR), Ridge Linear Regressor (RLR), Random Forest Regressor (RFR), and Multi-Layer Perceptron (MLP), were tested at both 50 and 250 m depth during the dataset preparation phase. Results for 250 m are shown in Figure 2. Based on these results, the multi-output RFR was selected as the most suitable algorithm for this multi-input/multi-output problem. Results are also analyzed using the built-in feature importance, which is based on the reduction in variance.

## 3   Performance of the Random Forest Regressor

Several dataset combinations were used as inputs for the RFR (described on section 2.3). For all the combinations, the RFR was trained on 80% of the data, with the remaining 20% set aside for testing. Experiments were conducted in the NA and STG

**Table 2.** Matrix sizes for the different datasets and depths. Dimensions specified as: $samples \times features \times outputs$

| Depth | Region | GCGOBGC | S3OLCIBGC | S3IOPs | S3OLCI |
|-------|--------|---------|-----------|--------|--------|
| **50 m** | North Atlantic | $389 \times 32 \times 26$ | $389 \times 37 \times 26$ | $389 \times 26 \times 26$ | $389 \times 15 \times 26$ |
| | Subtropical Gyres | $324 \times 32 \times 26$ | $324 \times 37 \times 26$ | $324 \times 26 \times 26$ | $324 \times 15 \times 26$ |
| **250 m** | North Atlantic | $389 \times 32 \times 126$ | $389 \times 37 \times 126$ | $389 \times 26 \times 126$ | $389 \times 15 \times 126$ |
| | Subtropical Gyres | $324 \times 32 \times 126$ | $324 \times 37 \times 126$ | $324 \times 26 \times 126$ | $324 \times 15 \times 126$ |

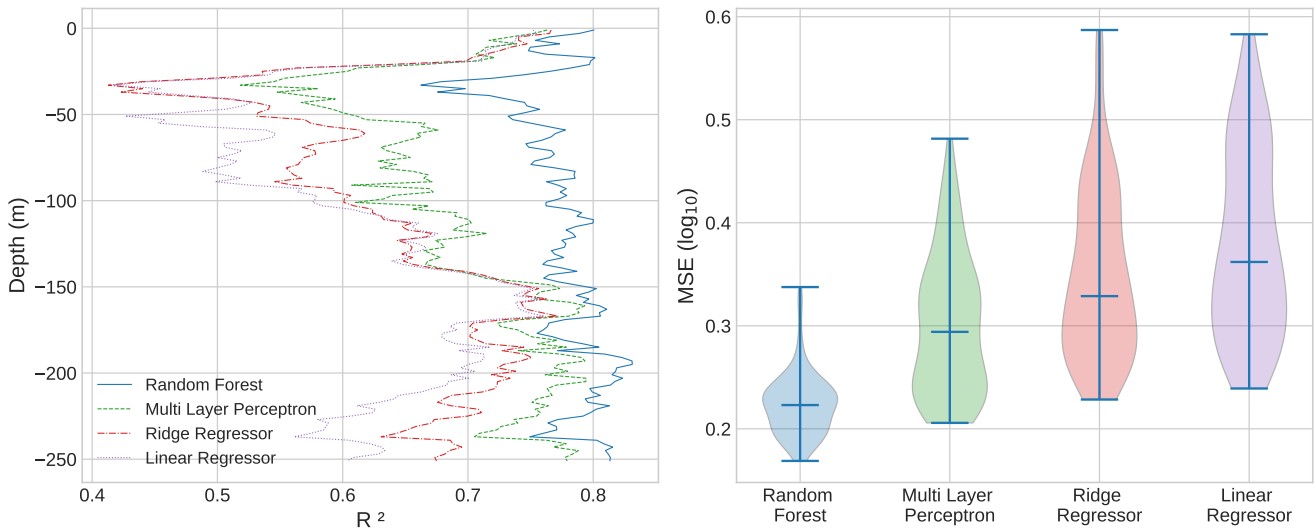

**Figure 2.** Comparison of different multi-output regression models for estimating vertical profiles of $b_{bp}$ up to 250 meters depth. Left: Depth-resolved $R^2$ values for four regression models: Random Forest, Multi-Layer Perceptron, Ridge Regressor, and Linear Regressor. Right: Violin plots of the Mean Squared Error (MSE, $\log_{10}$-transformed) distributions for each model.

regions for 26 layers in the range 0-50 m and for 126 layers in the range 0-250 m. The test dataset was exclusively used to evaluate model performance and was never exposed to the regressor during training. For each regression model, we analyzed the key features that contributed to improving the estimation of $b_{bp}$ in the different combinations. The models were finally validated using two independent floats in the NA and STG regions.

### 3.1 S3OLCIBGC: results with BGC-Argo and OLCI data

The equivalent set of data from the SOCA2020 experiment (GCGOBGC) is included in the statistical analysis to facilitate a comparison between our findings and previous studies. Table 1 and Table 2 present the input features and matrix sizes for the different experiments. In the following sections, we analyze the results of the RFR model applied to these datasets, starting with the GCGOBGC and S3OLCIBGC datasets to establish a baseline. In the NA region, 311 data points are used for training and 78 for testing, while in the STG region, 259 data points are used for training and 65 for testing. The results represent 20% of the dataset used for model testing.

### 3.1.1 Shallow waters: from 0 to 50 meters depth

The performance of the models trained to estimate $b_{bp}$ in the upper 50 meters of the water column are summarized in Figure 3 and Table 3. Figure 3 (A) includes depth-resolved $R^2$, Mean Absolute Error (MAE) and model bias, while panels (B) and (C)

**Table 3.** Statistics by region at 50 m and 250 m depth models with satellite and BGC-Argo. Median Absolute Percentage Deviation (MAPD) is expressed in % and Mean Absolute Error (MAE) in $m^{-1}$.

| Depth | Region | | GCGOBGC | S3OLCIBGC | S3IOPs | S3OLCI |
|---|---|---|---|---|---|---|
| **50 m** | North Atlantic | $R^2$ | 0.72 | 0.78 | 0.74 | 0.77 |
| | | MAPD | 8.19 | 10.77 | 13.46 | 12.96 |
| | | MAE ($\times 10^{-4}$) | 3.11 | 2.86 | 3.04 | 2.89 |
| | Subtropical Gyres | $R^2$ | 0.87 | 0.86 | 0.88 | 0.84 |
| | | MAPD | 5.60 | 5.54 | 5.61 | 5.56 |
| | | MAE ($\times 10^{-5}$) | 4.16 | 4.50 | 4.39 | 4.81 |
| **250 m** | North Atlantic | $R^2$ | 0.84 | 0.81 | 0.80 | 0.80 |
| | | MAPD | 3.37 | 5.24 | 6.38 | 6.18 |
| | | MAE ($\times 10^{-4}$) | 0.85 | 1.02 | 1.12 | 1.09 |
| | Subtropical Gyres | $R^2$ | 0.90 | 0.89 | 0.88 | 0.88 |
| | | MAPD | 4.97 | 5.36 | 5.98 | 5.47 |
| | | MAE ($\times 10^{-5}$) | 3.19 | 3.46 | 3.74 | 3.74 |

show measured and predicted $b_{bp}$ profiles, along with relative error distributions for the NA and the STG, respectively. Feature importances for both regions and models (GCGOBGC and S3OLCIBGC) are presented in Figure 4.

In the NA region (green lines), the S3OLCIBGC model achieves a higher average $R^2$ (0.78) compared to the GCGOBGC model (0.72). MAE is also lower (2.86 vs. $3.11 \times 10^{-4}\,m^{-1}$). In the depth-resolved metrics, the S3OLCIBGC model performs better both at superficial and deeper layers, maintaining a relatively stable performance down to approximately 20 m. Below this depth, accuracy decreases, particularly across and beneath the average MLD (36 m), where vertical gradients in temperature, salinity, and density intensify, and $b_{bp}$ variability increases (see Figure 1). While the MLD is inherently dynamic and varies throughout the year, this depth represents a critical boundary in our observations, marking a clear threshold where the behaviour of the model diverges. The GCGOBGC model shifts from overestimating $b_{bp}$ in the upper 15-20 m to underestimating values at greater depths. In contrast, the S3OLCIBGC model exhibits a relatively constant negative bias near the surface ($1 \times 10^{-4}$ $m^{-1}$), which increases gradually with depth indicating a degradation in the performance of the model. The more balanced contribution of surface and subsurface features in S3OLCIBGC enables the model to better resolve the vertical variability in $b_{bp}$ below the optical depth. In contrast, the GCGOBGC model presents lower $R^2$, higher MAE, and lower MAPD values, all of which indicate a reduced capacity to capture the vertical $b_{bp}$ variability. These differences likely reflect the superior spatio-temporal fidelity of the S3OLCI match-ups ($\pm 1$ day, 300 m pixels), which enable tighter temporal and spatial coupling between satellite and float observations, compared to the broader $\pm 5$ day, 4 km of the GlobColour dataset.

In the STG region (blue lines), both models achieve higher performance than in the NA, reflecting the lower variability and more stable vertical structure of $b_{bp}$ in these oligotrophic waters. S3OLCIBGC and GCGOBGC models obtain similar results,

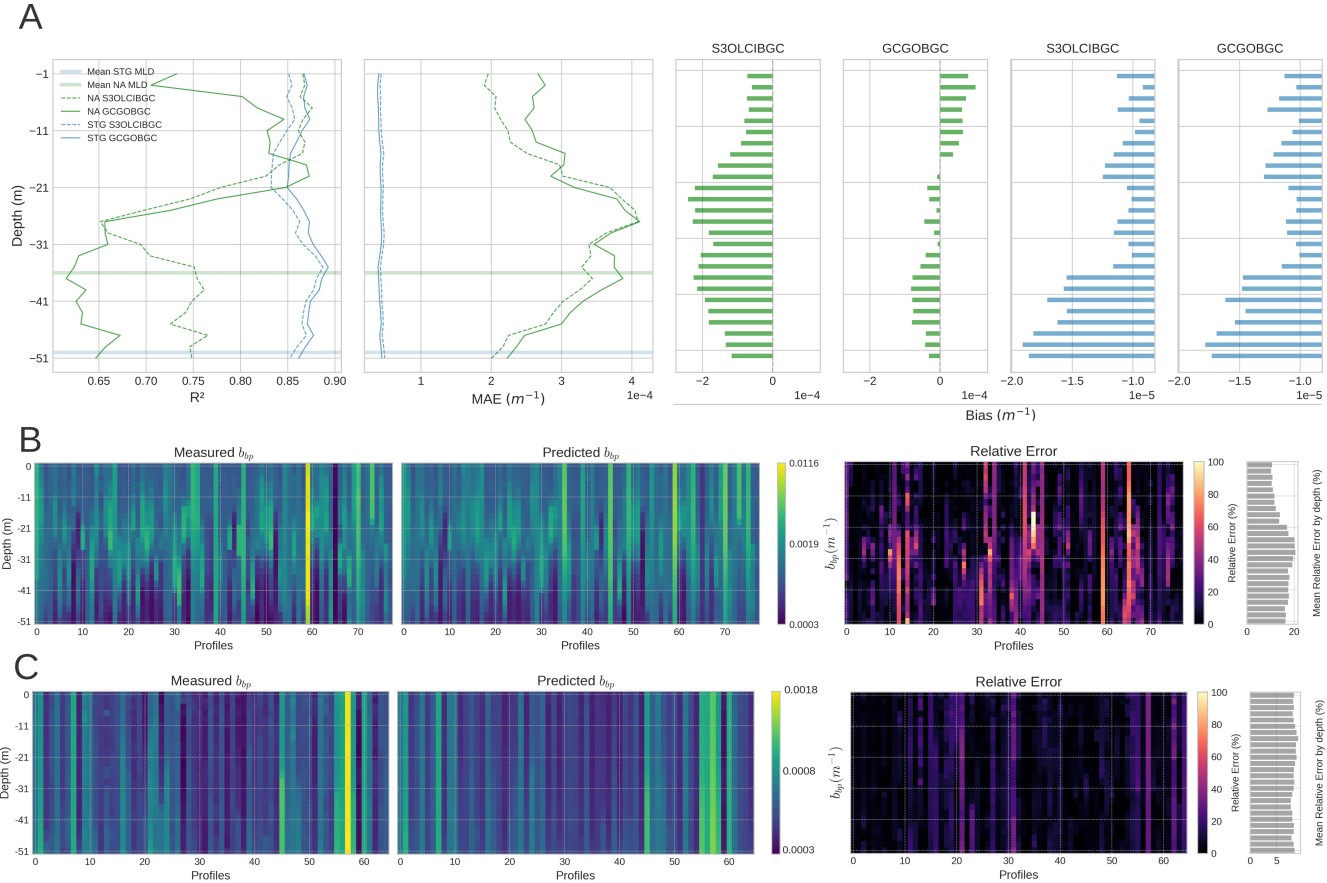

**Figure 3.** Model performance for estimating shallow water $b_{bp}$ profiles (0–50 m). **(A)** Depth-resolved metrics comparing model predictions using S3OLCIBGC and GCGOBGC sets as inputs: Coefficient of determination ($R^2$), Mean Absolute Error (MAE), and bias. Shaded horizontal lines indicate the average Mixed Layer Depth (MLD) per region. **(B -C)** Measured and predicted $b_{bp}$ profiles in NA (B) and STG (C) using S3OLCIBGC. The rightmost bars show the mean relative error by depth.

with mean $R^2$ values of 0.86 and 0.87 and MAE of 4.50 and 4.16 $\times 10^{-5}$ (m$^{-1}$), respectively. The depth-resolved metrics (Figure 3) show that models perform consistently throughout the upper 50 meters, with no marked degradation in $R^2$ or MAE near the average MLD ($\sim$ 50 m). In both cases, bias remains low in the upper 30 meters. However, starting around 35 m there is an increase in the bias, reaching their maximum near the bottom of the profile.

The feature importance analysis (Figure 4) shows that latitude is the most relevant feature in this case. This reflects that bio-optical conditions in the STG are very similar throughout the year; the day of year (doy) is less critical because of the low seasonality in these areas (Mignot et al., 2014; Cornec et al., 2021). The importance of the density and salinity features (*Dens_pc1* and *Sal_pc1*) reflects the barotropic dynamics of these oceanic regions, where isobars and isopycnals are stratified parallel to the ocean surface and vary together as depth is gained (Leonelli et al., 2022).

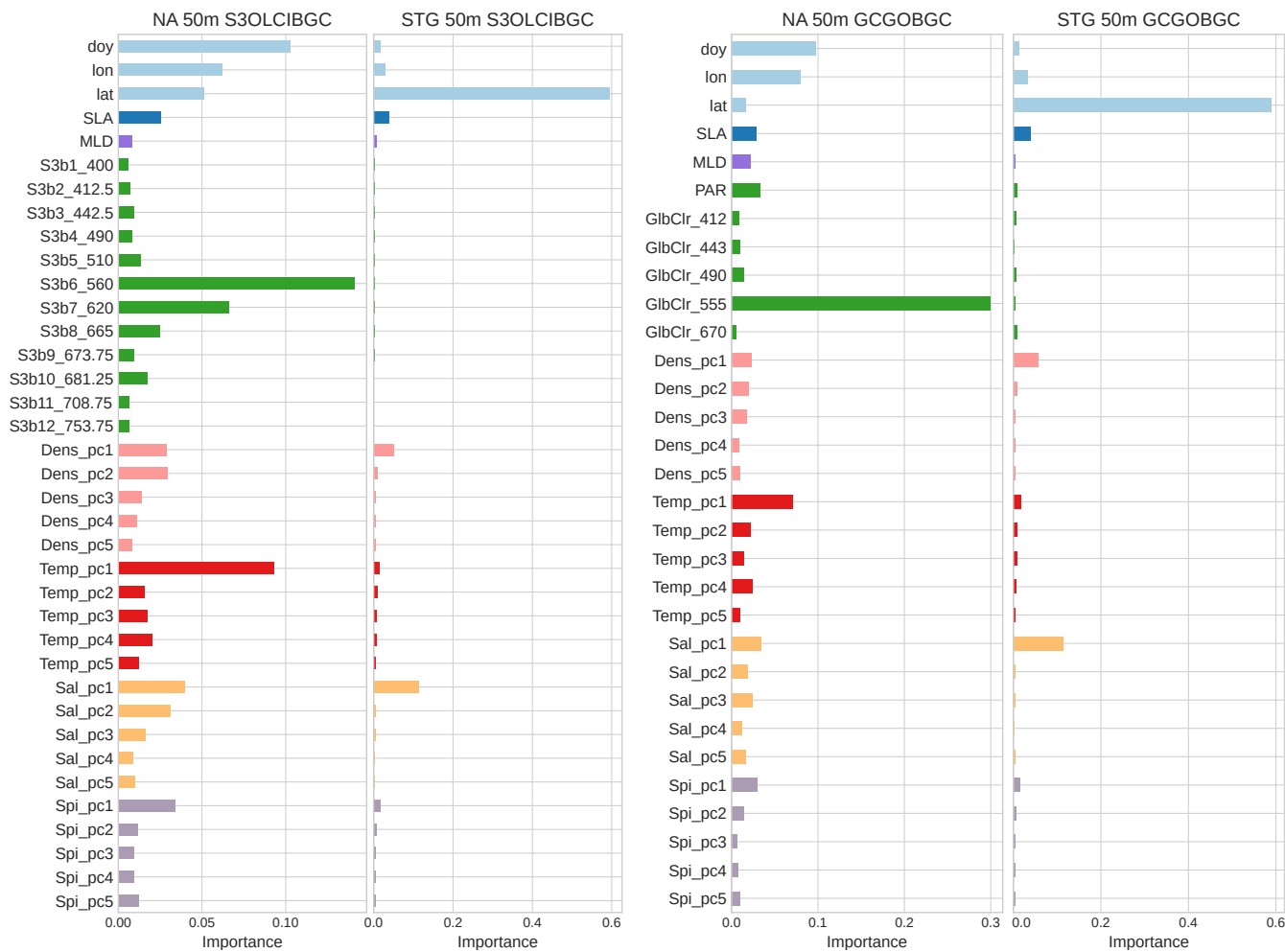

**Figure 4.** Feature importance (Gini importance) for the models trained with S3OLCIBGC and GCGOBGC to estimate $b_{bp}$ in shallow waters (0–50 m) in the NA and STG. Features are grouped by category according to color: (1) blue represent spatial and temporal descriptors, including day of year (doy), latitude, and longitude; (2) dark blue represents sea level anomaly (SLA); (3) purple indicates Mixed Layer Depth (MLD); (4) green corresponds to satellite reflectance bands from either Sentinel-3 OLCI or GlobColour with their central wavelengths; and (5) pink, red, orange, and light purple correspond to the first five principal components (PCs) derived from BGC-Argo profiles of density, temperature, salinity, and spiciness, respectively.

### 3.1.2 Deep waters: from 0 to 250 meters depth

The performance of the models to estimate $b_{bp}$ down to 250 meters is summarized in Figure 5 and Table 3. The S3OLCIBGC and GCGOBGC models obtain an $R^2$ of 0.81 and 0.84, respectively, with MAPDs of 5.24 and 3.37% and MAEs of 1.02 and 0.85 $\times 10^{-4}$ (m$^{-1}$). Shallower layers have larger errors in both models and correlates with the observed variability of $b_{bp}$ with

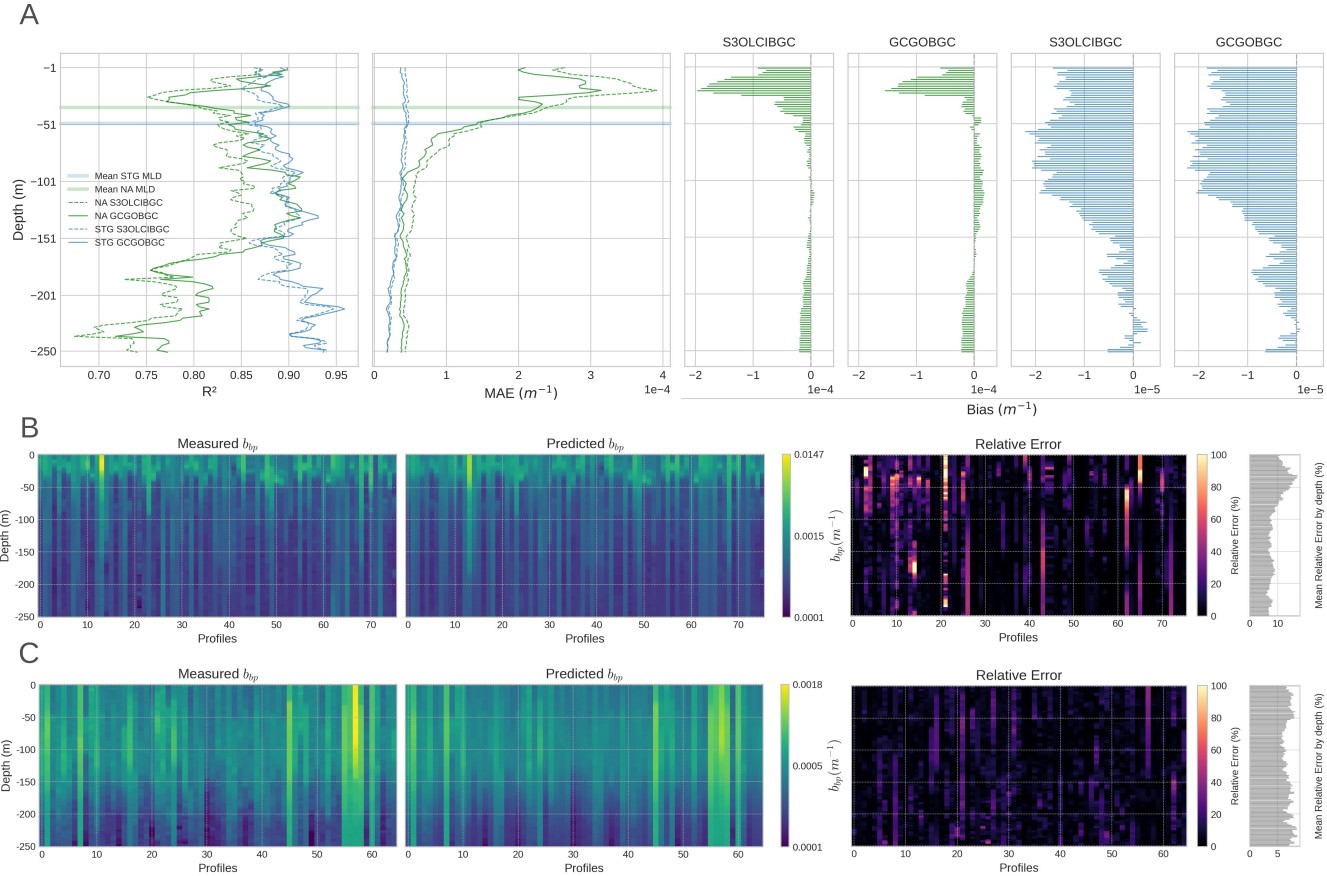

**Figure 5.** Model performance for estimating deep water $b_{bp}$ profiles (0–250 m). **(A)** Depth-resolved metrics comparing model predictions using S3OLCIBGC and GCGOBGC as inputs: Coefficient of determination ($R^2$), Mean Absolute Error (MAE), and bias. Shaded horizontal lines indicate the average Mixed Layer Depth (MLD) per region. **(B -C)** Measured and predicted $b_{bp}$ profiles in NA (B) and STG (C) using S3OLCIBGC. The rightmost bars show the mean relative error by depth.

depth in each region. The GCGOBGC does not experience overestimation in the most superficial layers, as was the case for
285 the 50 m models.

Feature importance analysis for both models (Figure 6) are similar to 50 m models and also highlights the dominant role of the first principal components (*pc1*) derived from BGC-Argo physical variables (density, temperature, salinity, and spiciness). It aligns with the correlation heatmap (Figure 6 right), which illustrates how these first PCs correlate with $b_{bp}$ across depth, showing a strong positive correlation ($> 0.6$) in the 180–250 m range. In these deeper layers, where biogeochemical processes 290 such as particle sinking, remineralization, and carbon export are more active, the relationship between physical stratification and $b_{bp}$ becomes stronger and more linear.

In the STG region, model performance exceeds that of the North Atlantic, likely due to the more optically homogeneous conditions despite the strongly stratified nature of these oligotrophic waters (Figure 5 (C)). Similar patterns have been observed in stratified waters, where optical properties like beam attenuation remain relatively homogeneous (Kitchen and Zaneveld, 1990). While the average MLD is around 50 meters, no significant increase in error is observed until approximately 120 meters. This deeper threshold aligns with the region typically associated with the Deep Biomass Maxima (DBM), which forms at the interface between the nutrient-depleted surface layer and the light-limited mesopelagic zone (Cornec et al., 2021). In the STG, this transition zone, often located between 150 and 200 meters (Mignot et al., 2014), appears as a boundary where the predictive skill begins to slightly decline—reflected in the gradual increase in relative error and a subtle shift in bias profiles. Feature importance for the STG 250 m models is not shown, as it is similar to the obtained in the 50 m models. In both cases, latitude emerges as the most relevant predictor.

## 3.2 S3OLCI: results of Sentinel-3 OLCI without BGC-ARGO data

As demonstrated in the previous experiments, satellite-derived features play a significant role in the models when profile depths reach 50 meters, thus answering the initial hypothesis of this study. It is clear that sea surface signals help to estimate $b_{bp}$ at subsurface levels. However, the extent of this contribution across different depth layers became evident only when comparing models trained with different depth limits. The feature importance of the 50 m depth models shows that, at least in the NA region, the parameters measured by satellite sensors are just as relevant as the inputs from the floats. For this reason, we carried out a last experiment with only satellite data (S3OLCI) to check how the models perform in-depth with the normalized water-leaving reflectance bands. Additionally, we investigated the contribution of satellite-derived IOPs from the C2RCC processor, that is, adding the absorption and scattering variables as input features (S3IOPs).

In the NA region, the model using only reflectance data (S3OLCI) outperforms the model that includes both reflectance and the absorption and scattering (S3IOPs) (see Table 3 and Figure 8 (A)). While the MLD is still a barrier, accuracy improves beyond this depth for approximately another 10 meters. In the $b_{bp}$ profiles (Figure 8 (B)), despite the errors noted in deeper estimations, the model is capable of predicting significant contrast events using only surface data from 36 meters onward, except in a specific case characterized by high $b_{bp}$ values (profile 59). In the feature importance ranking (Figure 7), the 620 nm band is the most relevant of the spectrum. However, the spatio-temporal features (day of year, longitude and latitude) seem to have greater weight than the results obtained with the data sets that include BGC-Argo data at the same depth (see Section 3.2.1).

In the STG region, the S3IOPs model achieves better results (Table 3). However, it is possible to see how the model is not able to predict some spikes along the water column (Figure8 (C)). In the feature importance ranking, latitude remains the most relevant feature. The improved performance of the S3IOPs model, compared to the S3OLCI (reflectance-only) model, could be attributed to the contribution of marine particle scattering at 443 nm (*iop_bpart*) provided by the C2RCC processor.

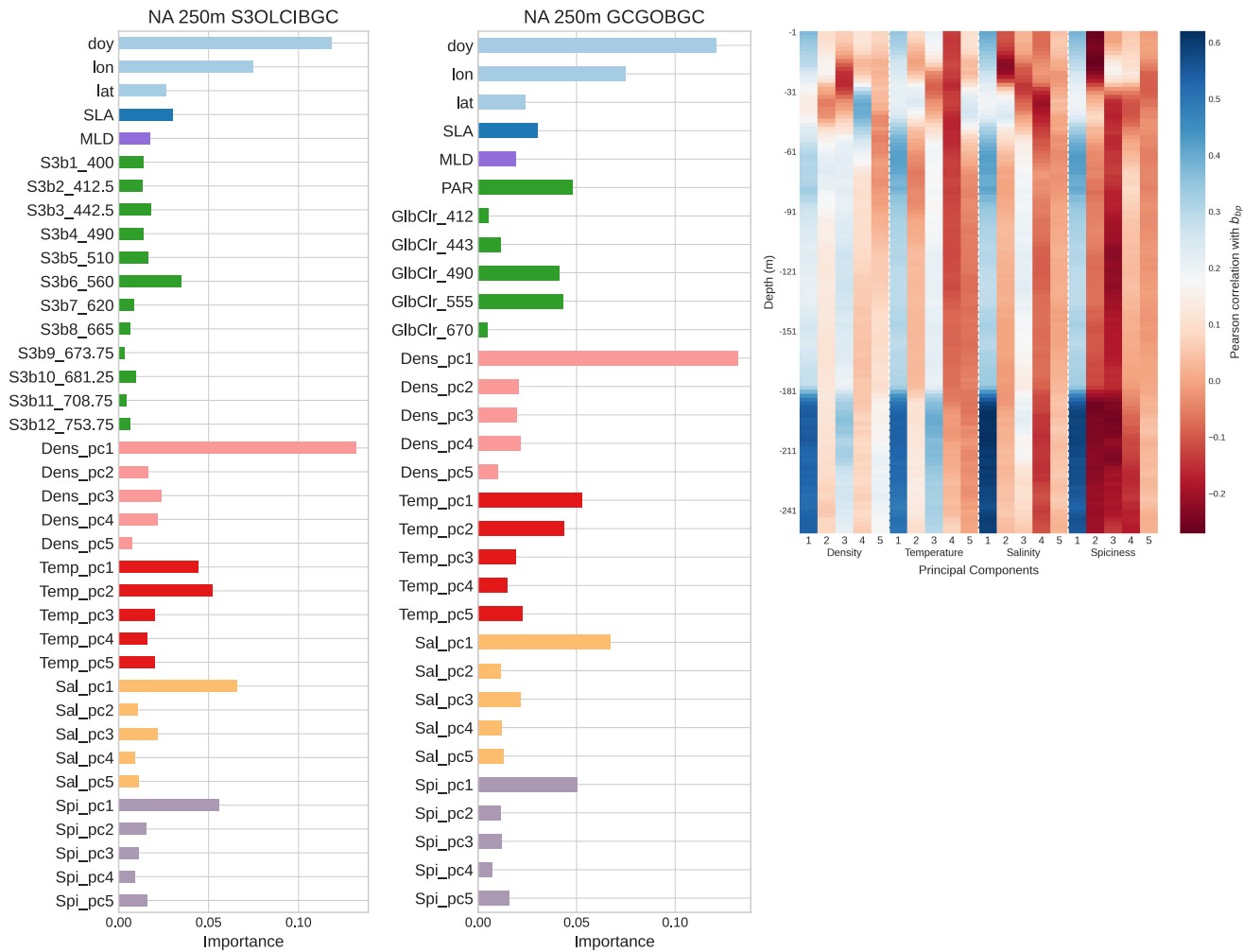

**Figure 6.** Left: Feature importance of the S3OLCIBGC and GCGOBGC models for $b_{bp}$ estimation down to 250 m in the North Atlantic. Right: Correlation Matrix between $b_{bp}$ and the first five PCs of each BGC-Argo physical variable as a function of depth.

### 3.3 Validation with Independent Floats

The previously trained RFR models are applied to predict $b_{bp}$ values using independent float data that is not included in the training or testing sets. Statistical metrics and corresponding scatter plots are provided in Table 4 and Figure 9.

In the NA region, the float identified as WMO 6902545 (see location in Figure 1) yields better estimates with the S3OLCI models ($R^2$ ranging from 0.41 to 0.44) compared to the reference GCGOBGC model, where the $R^2$ value drops to 0.26. This improvement is also visible in the absolute and relative error estimations (MAE, RMSE). Figure 9 (A) reveals the higher $b_{bp}$ variability along the water depth in the NA region, as indicated by the colour scale. There is an overestimation in the surface

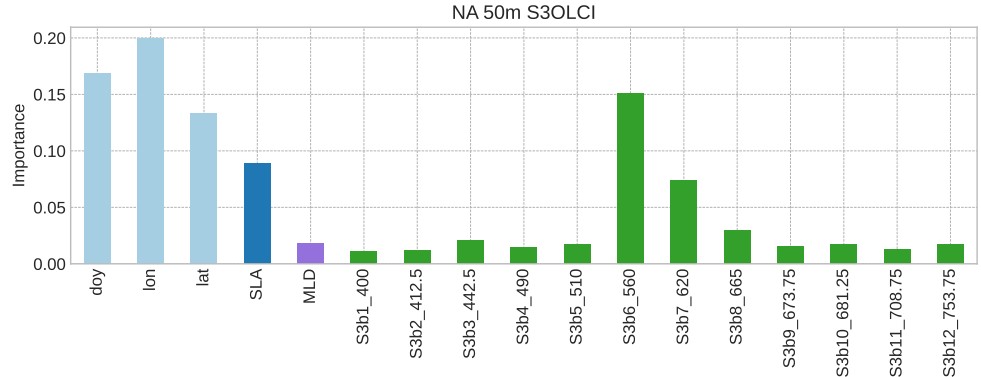

**Figure 7.** Feature importance of the S3OLCI model for $b_{bp}$ estimation down to 50 m in the NA area.

**Table 4.** Validation with independent floats by region at 50 m and 250 m depth models with satellite and BGC-Argo data. MAPD is expressed in %, MSE and RMSE in $m^{-1}$.

| Depth | Region | | GCGOBGC | S3OLCIBGC | S3IOPs | S3OLCI |
|-------|--------|--------|---------|-----------|--------|--------|
| **50 m** | North Atlantic | $R^2$ | 0.26 | 0.44 | 0.44 | 0.41 |
| | | MAPD | 38.48 | 33.74 | 27.96 | 32.5 |
| | | MAE $\times 10^{-4}$ | 7.00 | 5.91 | 5.61 | 5.67 |
| | | RMSE $\times 10^{-4}$ | 9.05 | 7.38 | 7.05 | 7.13 |
| | Subtropical Gyres | $R^2$ | 0.65 | 0.63 | 0.64 | 0.63 |
| | | MAPD | 5.06 | 4.90 | 5.69 | 5.99 |
| | | MAE $\times 10^{-5}$ | 2.93 | 3.08 | 3.09 | 3.12 |
| | | RMSE $\times 10^{-5}$ | 3.60 | 3.73 | 3.67 | 3.73 |
| **250 m** | North Atlantic | $R^2$ | 0.32 | 0.31 | 0.29 | 0.29 |
| | | MAPD | 6.56 | 6.38 | 6.41 | 6.37 |
| | | MAE $\times 10^{-4}$ | 7.50 | 7.68 | 7.44 | 7.47 |
| | | RMSE $\times 10^{-4}$ | 10.4 | 10.5 | 10.6 | 10.7 |
| | Subtropical Gyres | $R^2$ | 0.58 | 0.53 | 0.54 | 0.56 |
| | | MAPD | 7.38 | 8.34 | 8.28 | 8.06 |
| | | MAE $\times 10^{-5}$ | 3.88 | 4.19 | 4.18 | 4.02 |
| | | RMSE $\times 10^{-5}$ | 4.92 | 5.32 | 5.34 | 5.13 |

measurements (less than 30 m) and a slight underestimation at deeper depths. This validation set includes data from several dates in 2017 and 2018, spanning from April to August. These temporal variations explain some of the observed drifts in the plots, where different float cycles (water depth profiles) are also evident.

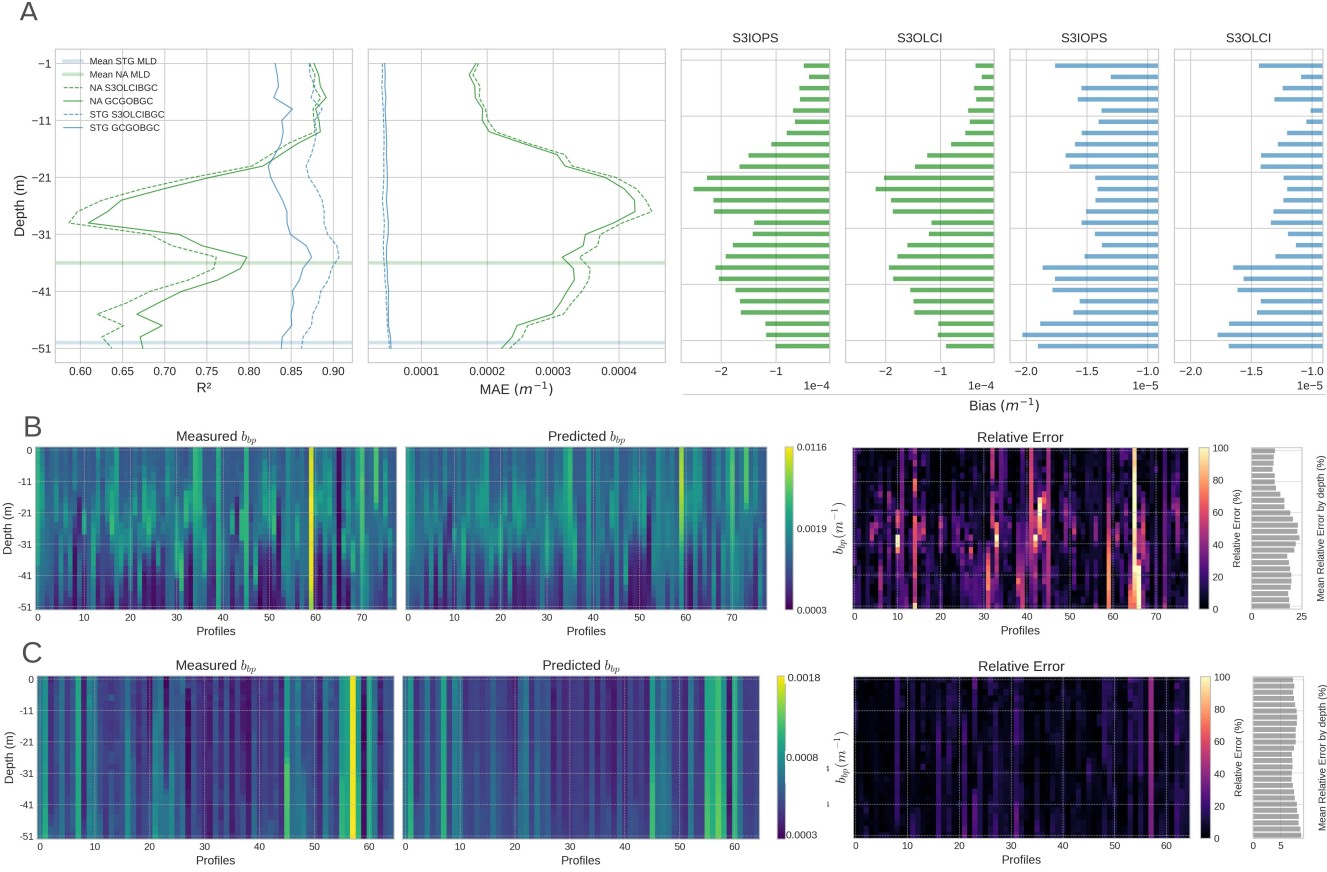

**Figure 8.** Model performance for estimating shallow water $b_{bp}$ profiles (0–50 m) with only satellite data. **(A)** Depth-resolved metrics comparing model predictions using S3OLCI and S3IOPS as inputs: Coefficient of determination ($R^2$), Mean Absolute Error (MAE), and bias. Shaded horizontal lines indicate the average Mixed Layer Depth (MLD) per region. **(B-C)** Measured and predicted $b_{bp}$ profiles in NA (B) and STG (C) using S3OLCI. The rightmost bars show the mean relative error by depth.

The STG statistics and plots for the float with identifier 3902125 show better correlation coefficients and lower errors compared with the NA. The datasets incorporating S3OLCI data yielded the best results. In Figure 9 (B), two clusters of data are visible: one associated with low $b_{bp}$ values and the other clustering around slightly higher values. The models tend to underestimate the lower $b_{bp}$ values, while the higher values show a closer fit to the 1:1 line. However, in the model that uses only reflectance data (S3OLCI) a clear overestimation of higher values occurs. Unlike in the North Atlantic region, depth separation is not evident here, but the lower values correspond to measurements taken during the winter months in the South Atlantic Gyre, while the higher values were recorded during the summer months in the Southern Hemisphere, where the float was located. These results reinforce the observations made in previous sections: models provide more accurate $b_{bp}$ estimations in the STG region than in the NA, confirming the effectiveness of using the S3OLCI bands and derived C2RCC IOPs at shallow water depths.

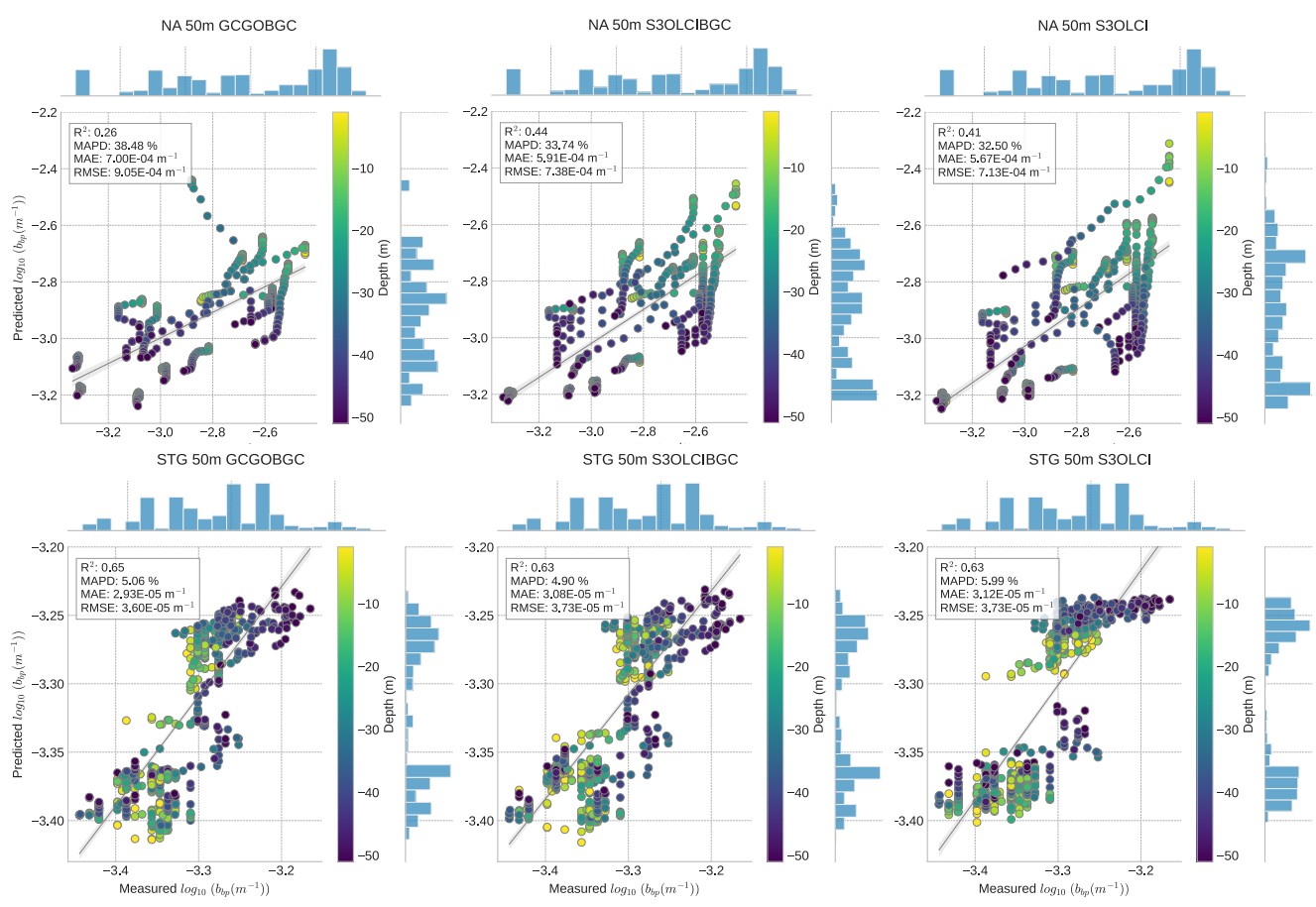

**Figure 9.** (A) Scatter plots with marginal histograms with the validation of the 50 m model performance on an independent float on the NA region (ID 6902545) and (B) on the STG region (ID 3902125). The color scales shows the depth of the measurements and $b_{bp}$ values are in log10.

## 4 Discussion and Conclusion

Previous studies estimating $b_{bp}$ from satellite-derived remote sensing reflectance ($R_{rs}$) have typically employed traditional

statistical approaches mostly focused on surface layers. In Bisson et al. (2019), $b_{bp}$ profiles from floats were processed by averaging $b_{bp}$ values within the surface mixed layer, followed by a comparison between different sensors and $b_{bp}$ retrieval inversion products by NASA. In that case study, OLCI -with data from Reduced Resolution mode at 1.2 km pixel resolution-underperformed compared to MODIS (Moderate Resolution Imaging Spectroradiometer) with 1 km at nadir (r= 0.32 to 0.47 and r = 0.60 to 0.79 respectively). This difference was attributed to higher coefficients of variation (30% for OLCI and 5%

for MODIS) across bands between 412 and 555 nm and aerosol optical thickness at 865 nm. In the present work, OLCI Full

Resolution (FR) data, with a spatial resolution of 300 m, is used. Additionally, the most relevant wavelength in some of our models (620 nm) was not considered in Bisson et al. (2019).

In the present study, vertical estimates of the particulate backscattering coefficient ($b_{bp}$) are calculated along the water column. We have applied a multi-output Random Forest model, which shows promising results, especially within the first 50 meters in the Subtropical Gyres. However, in dynamic regions such as the North Atlantic, the results are less consistent, suggesting that further research is needed to understand how the complexity of the physical state of the water column modifies the $b_{bp}$ vertical fluxes. Nevertheless, the focus of our work is on the analysis of the contribution of satellite-derived water-leaving reflectance to $b_{bp}$ estimation within the first 250 meters.

In deeper layers, where biogeochemical processes like particle sinking, remineralization, and carbon export are more pronounced, the relationship between physical stratification and $b_{bp}$ seems to become stronger and more linear. In contrast, the correlation between these variables is weaker —though still positive— in the upper layers (0–30 m) and below the mixed layer depth (61–91 m). Density and temperature can offer additional insights into the upper water column, helping to pinpoint processes associated with the depth and intensity of the pycnocline or the MLD. Collectively, these physical features could allow the model to infer the transfer of $b_{bp}$ from the sunlit surface to the twilight zone by learning from stratification patterns.

Satellite features have proved to be indeed relevant for the $b_{bp}$ estimations, especially in the Subtropical Gyres region, as mentioned. These waters, characterized by high stratification, rely heavily on nutrient injection from deeper zones, as the upper euphotic zone is typically nutrient-limited. In fact, Letelier et al. (2004) and Mignot et al. (2014) describe these gyres as a two-layer system: an upper layer nutrient limited but not light-limited, and a deeper layer that is light-limited but has greater nutrient access. These authors also highlight a seasonal distinction, with winter bringing greater water mixing than summer. During winter, average light intensity for PAR in the mixed layer decreases while turbulence increases. This seasonal variation may explain the two distinct clusters observed in the validation exercise for the STG region, since two clusters of data are observed, one belonging to the winter of 2017, with slightly higher values; and the second one coincident with the spring-summer 2018.

The inclusion of satellite surface data, along with derived parameters such as inherent optical properties (IOPs), in combination with *in situ* profile data, should be considered for estimating $b_{bp}$, and by extension, approximating particulate organic carbon (POC), at least for layers up to 250 meters depth. It is important to note that organic carbon fixation primarily occurs in the upper ocean layers. This organic matter is subsequently transformed through respiration, particle aggregation, zooplankton grazing, feces production, and microbial decomposition (Siegel et al., 2014), before a fraction of it sinks to deeper layers.

The models that relied exclusively on satellite data (S3OLCI and S3IOPs) produced reasonable estimations for the upper layers in both the North Atlantic and Subtropical Gyres regions. This is encouraging, as satellite data, with its synoptic spatial coverage can efficiently complement Argo float measurements. Satellite observations provide valuable insights into mesoscale ocean processes over various temporal ranges, extending at least the past three decades. Since remote sensing products can only reach at about 20% of the euphotic zone, the importance of extending surface observations to deeper layers using autonomous floats or other devices is critical (Claustre et al., 2010).

Future work should be focused on enlarging the database with new BGC-Argo profiles and satellite data, extending the study to new areas of the global ocean. Some details that could enrich the analysis is the role of the MLD on the different regions in

order to further understand the effect that it has on biochemical parameter estimations. Sensors with extended capabilities, like the hyperspectral NASA PACE, might be also a path of research to follow, since we have seen that adding new wavelengths had a positive effect on the results of our models compared with sensors with less capabilities. Possible improvements in the detection of CDOM with the UV bands can be an important contribution to better estimating particulate organic material (POM)

and, consequently, POC. It has been determined that there is an increase in photoproduction of $CO_2$ from CDOM (Bélanger et al., 2006) due to the increase in UV radiation and the decrease in sea ice because the risen of global temperatures. Organic carbon is separated into particulate and dissolved organic carbon (DOC). There is a potential use of aCDOM to improve DOC estimations (especially in coastal waters) together with physical variables like sea surface temperature or salinity. If CDOM can really improve DOC estimations and we can do it globally with satellites, a better understanding of the relationship between

DOC and POC could also be analyzed temporally and spatially.

*Data availability.* Both BGC-Argo measurements and OLCI data are open and freely available for the scientific and public community. Python scripts with the model will be available from GitHub on the ISP site in accordance with our group policy of publishing developed models in open access: https://github.com/IPL-UV

*Author contributions.* JGJ, JAL, and ABR designed the experiments and prepared the match-up data set based on the BGC-Argo data pre-

processed by RS. JGJ processed the data and run the models. ABR made the independent validation. All four author contributed with the revision and writing of the paper.

*Competing interests.* No competing interests are present

*Acknowledgements.* AI4CS - GVA PROMETEO Projecte "Artificial Intelligence for complex systems: Brain, Earth, Climate, Society", funded by Conselleria de Innovación, Universidades, Ciencia y Sociedad Digital, CIPROM/2021/56

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
