# Peer review of "Combining BGC-Argo floats and satellite observations for water column estimations of particulate backscattering coefficient"

_EGUsphere, 2024_

## Referee Comment (RC2)

[referee-annotated manuscript omitted]

---

## Author Comment (AC1)

Dear Reviewer, we sincerely appreciate your valuable review and feedback on our manuscript. We agree that it is important to emphasize the key contributions and innovations of our study. In response to your comments, we have rewritten the abstract and discussion sections to clearly summarize our findings. Additionally, we have explicitly highlighted the main contributions of our work at the end of the introduction to enhance clarity and impact. A new version will be uploaded into the system when permitted. Some excerpts from the text have been copied into the responses for clarity.

The current manuscript builds upon existing research to explore the use of multi-output random forest models for retrieving backscattering coefficient (bbp) data at various depths using different input datasets. These inputs include enhancements in spatial resolution and a diversity of data types. However, the draft in its present form is somewhat rudimentary. The work presented is not effectively summarized in the abstract and discussion sections, and the highlights and innovations of the study are not prominently featured. I recommend that the authors address these issues by clearly outlining the study's contributions and innovations at the end of the introduction.

Below are specific suggestions for improvement:

1. The introduction initially mentions Particulate Organic Carbon (POC) using its abbreviation without first presenting its full name and explaining its significance within the study's context. This may confuse readers unfamiliar with the term.

   The full name and abbreviation were first introduced in the abstract but are now also included in the introduction of the new version.

   Additionally, the discussion on why profiling POC is challenging is insufficiently developed. A more detailed explanation of these measurement difficulties is necessary to establish the research problem's significance and to clearly justify the study's objectives. Providing a comprehensive background on POC and elaborating on the challenges in measuring it will better prepare readers for the research presented.

   Thanks for pointing this out. We have modified the introduction and added a more detailed description in the new version of the manuscript (see in question number 3).

2. The second paragraph of the introduction discusses Apparent Optical Properties (AOP), which does not appear to be directly related to the paper's main focus. This section may detract from the introduction's clarity and coherence by introducing a topic that is not central to the study's objectives. It is important to ensure that the introduction remains focused on the key themes and research questions. If AOP is not essential to the main argument, consider removing this section or significantly condensing it to maintain the introduction's focus and engage readers with the paper's central themes.

   One of the aims of the manuscript is to evaluate if the Inherent Optical Properties (IOP), which are provided as a derived product from the Sentinel-3 OLCI-C2RCC data, can improve the retrieval of bpp. This product is directly related to AOP and a comprehensive background is provided in the introduction for this reason. However, we have rewritten this paragraph to highlight why it is important in this study.

3.  The logical flow between the introduction's first two paragraphs is somewhat disjointed, potentially hindering the reader's understanding of the paper's overall direction. Furthermore, the latter paragraphs lack a detailed analysis of the current research landscape. The discussion of existing studies is limited and does not clearly identify the knowledge gaps this paper aims to address. To enhance the introduction, revise the first two paragraphs to improve their logical structure and coherence. Additionally, include a more comprehensive review of the current research, highlighting specific gaps in the literature and the problems this study seeks to solve. Incorporating more examples of relevant previous research will strengthen the context and rationale for the study, providing a clearer foundation for the paper's contributions.

Thank you for your comments. We have rewritten the introduction accordingly to clarify and highlight the objectives and novelties of this study.

"*The ocean covers approximately 70\% of Earth's surface and plays a fundamental role in regulating climate dynamics. It redistributes energy and carbon through a variety of physical and biogeochemical processes. Among these processes, the biological carbon pump facilitates the transfer of $CO_{2}$ from the atmosphere to the ocean floor by enabling the production and sinking of particulate organic carbon (POC), which becomes sequestered in deep-ocean sediments. POC originates from living organic carbon, primarily produced by photosynthetic organisms such as phytoplankton, which thrive in the sunlit upper ocean layers. These organisms require carbon compounds, along with light and nutrients, to survive and reproduce(Falkowski et al., 1998; Siegel et al., 2014).Their presence and abundance reflects the interplay of resources and losses in the environment(Behrenfeld et al., 2006) with populations maintaining daily division cycles even in regions where nutrients appear to be depleted beyond detection limits (Ribalet et al., 2015; Vaulot and Marie, 1999). The observed populations represent a balance where new biomass produced each day is matched by consumption through grazing and other loss processes (Landry and Hassett, 1982; Calbet and Landry, 2004) maintaining relatively stable populations despite continuous growth and turnover. Quantifying phytoplankton biomass and carbon content is crucial to understanding these ecosystem dynamics and their role in carbon cycling. Traditionally, chlorophyll-a (chl-a}) concentration has been used as a proxy for phytoplankton biomass, but its interpretation is complicated due to physiological photoacclimation, which affects intracellular pigment content without necessarily indicating changes in biomass. The Particulate Backscattering Coefficient (bbp) has been recognized as a stable optical proxy for phytoplankton biomass and carbon content as it is sensitive to the abundance, size distribution, and composition of suspended particles, rather than pigment concentration alone (Behrenfeld and Boss, 2006; Graff et al., 2015; Martinez-Vicente et al., 2013). Unlike chl-\textit{a}, which can underestimate biomass in stratified and oligotrophic waters, bbp remains relatively unaffected by photoacclimation effects, making it particularly useful for studying carbon fluxes across different oceanic regions and depth layers. The complex interaction between key variables (usually non-linear) and limited sampling resolution in dynamic environments, combined with the technical challenges of depth-resolved measurements, contribute to gaps in our understanding of specific marine processes, such as carbon sequestration, nutrient cycling, sedimentation and the ocean-atmosphere CO2 exchange (...)*"

4. The introduction should underscore the importance of bbp in POC measurement, as well as the deficiencies and areas for improvement in current bbp products. While the introduction currently highlights the significance of POC, it does not adequately stress the critical role of bbp. Clarify whether POC estimation relies solely on bbp and discuss its specific importance in this context. Additionally, expand upon the current state of bbp data by discussing the limitations of existing bbp products and the shortcomings of related algorithms. For instance, accurately deducing inherent optical properties (IOPs) from apparent optical properties (AOPs) is crucial for POC retrieval models based on IOPs, but this process can be challenging. Furthermore, the complex optical conditions in coastal areas can lead to significant spatial heterogeneity in POC distribution, introducing uncertainty in POC estimation even when using advanced methods. Addressing these points will provide a clearer context for the study's objectives and the need for improved bbp products.

We have added a paragraph in the Introduction where we tackle the following points:

- Remark the bbp use as a proxy of POC
- How we can derive bbp: from in situ data using the BGC-Argo floats (sensors of backscatter at 700 nm); from satellite, where several algorithms have been developed like NASA's OBG group bbp_sat product.
- From satellites Rrs is used and an inversion model is applied to derive IOPs. The retrieval of bbp from satellites requires of a forward model that shows the relation between Rrs and bbp (i.e. GIOP, QAA and others).

It now reads:

*"The bbp parameter is an inherent optical property (IOP) of water, and it has been widely recognized as a robust bio-optical proxy for POC (Cetini´c et al., 2012; Sullivan et al., 2013). However,  b_bp measured by floats can have an uncertainty of the order of 10–15% (Bisson et al., 2019).  These uncertainties stem from the instrumental drift, the sensor calibration limitations, and the reliance on manufacturer calibration files rather than sensor-specific calibrations using dark counts. While autonomous platforms provide extensive spatial and temporal coverage, these factors must be considered when interpreting bio-optical datasets to ensure accuracy and reliability. IOPs are intrinsic characteristics of water, determined solely by its composition and are independent of the external light field or the geometrical angle conditions during observation. These properties include absorption, elastic scattering, inelastic processes (such as fluorescence and Raman scattering), and attenuation, which describe how light behaves and propagates through water. IOPs are essential in studying light interactions in aquatic environments, as they reflect the presence of dissolved organic matter, phytoplankton and suspended particles. The b_bp can be measured by autonomous platforms spread out across the ocean, such as the Biogeochemical-Argo (BGC-Argo) profiling floats(Claustre et al., 2020);  or estimated from onboard satellite sensors, such as the Sentinel-3 Ocean and Land Colour Instrument (EUMETSAT, 2019; Jorge et al., 2021; Koestner et al., 2024). Designing observational strategies based on*

*combining the two approaches constitutes a fundamental tool for improving knowledge of ocean processes.*"

The calculation of inherent optical properties (IOPs) or concentrations of water constituents from reflectances always involves some degree of uncertainty (IOCCG, 2019). Two primary factors contribute to this: 1) The inverse relationship between IOPs and water reflectance spectra is an underdetermined system. This means that the information contained in reflectance spectra is significantly lower than the number of variables influencing those spectra. As a result, different combinations of IOPs can produce nearly identical reflectance spectra, leading to inherent ambiguity between IOPs and remote sensing reflectance ($R_{rs}$). 2) The natural variability of all components that determine reflectance spectra is extremely high. This includes the optical properties of the atmosphere and water, their vertical distribution, and the characteristics of the air-sea interface. Because of this complexity, any retrieval algorithm simplifies the natural system and is only effective within the scope defined by its underlying optical model assumptions.

☐ The $b_{bp}$ from Argo floats is explained in detail in section 2.2.
☐ Derivation of the satellite IOPs from C2RCC is later explained in section 2.3

5. It is crucial to provide specific details about the data collected from each dataset, including the exact variables used, the time range of data collection, website links for accessing the data, and the dates when the data were accessed. Currently, Table 1 lacks sufficient information, and the time frames for the BGC-Argo data and other datasets are not clearly stated. To improve clarity and completeness, ensure that all necessary details are included in the data section, allowing readers to understand the scope and sources of the data used in this study.

*We have added the following text in the new version of the manuscript and changed figure 1.*

*"The temporal distribution of the match-ups shows a clear seasonal bias, with most data concentrated between May and September, particularly during 2017. This uneven distribution is primarily due to the limited availability of cloud-free satellite observations required to match with BGC-Argo profiles, especially during winter months when cloud cover and low solar angles reduce the quality of remote sensing products."*

[Figure]

*Figure 1. Global map showing the geographic locations of the BGC-Argo floats and satellite data matchups. [Botttom row] Temporal coverage of matchups by year (left) and month (middle) for the North Atlantic (NA, green) and Subtropical Gyres (STG, blue). Vertical profiles (right) of bbp from floats, where solid lines show mean values, shaded areas ±1 standard deviation, and the dashed red line the average Mixed Layer Depth (MLD).*

6. In the methods section, the use of Principal Component Analysis (PCA) for dimensionality reduction of high-dimensional features is mentioned, stating that "After this feature reduction on the high-dimensional variables, the 250 m and 50 m measurements with 126 and 26 inputs are reduced to 5 components for each variable, resulting in a total of 20 features. This method still retains 99% of the information." However, this section lacks supporting data and visualizations to illustrate the PCA results. To enhance clarity and effectiveness, include data tables or figures that demonstrate the specific components selected and their contributions to the overall variance. This will help readers better understand the impact of PCA on the feature set and validate the claim that 99% of the information is retained.

The application of PCA for dimensionality reduction is a well-established and widely used technique for reducing the dimensionality of large datasets while retaining the most important information. Given its common application in the field, we did not initially include a graphical representation of the PCA results in the manuscript. However, in response to your suggestion, we could include as supplementary material the PCA plot in the revised manuscript, although we consider that it does not add much to the analysis. PCA graph for all the variables and depth are shown below:

[Figure]

*Figure. Cumulative explained variance of the first six PCA components.*

7. In Section 2.5, the discussion is somewhat disorganized. The introduction of the Random Forest Regression model should precede the discussion of existing studies based on random forest models. Additionally, such content seems more appropriate for the introduction section, as it pertains to a review of existing research rather than the methods section. Moreover, the authors state, "All the previously mentioned algorithms, along with others such as Linear Regressor (LR), Ridge Linear Regressor (RLR), Random Forest Regressor (RFR), and Multi-Layer Perceptron (MLP), were tested for estimating bbp during the dataset preparation phase. Based on these results, the Random Forest Regressor (RFR) was selected as the most suitable algorithm for this multi-input/multi-output problem." Comparative results should also be presented to illustrate the differences in inversion results and the stability of various models. This will help substantiate the choice of the Random Forest Regressor as the most suitable algorithm for the problem at hand.

We have restructured this section according to the comments. Thank you for pointing it out. Now it reads like this:

*"There are two main approaches for dealing with multi-output regression problems. One way is to use univariate models, also known as problem transformation methods (Schmid et al., 2022; Borchani et al., 2015). These methods decompose the multi-output regression problem into multiple single-target problems, creating an independent model for each output. The predictions from these separate models are then combined. This approach ignores the relationships between the targets, which can adversely affect the prediction's overall accuracy. Alternatively, multivariate models are designed to capture dependencies and interactions between the outputs, potentially leading*

to more accurate predictions (Borchani et al., 2015). When and how to apply these two approaches depends on the nature of the data and the correlation between the targets. In our preprocessing results, PCA decomposition indicates a high covariance among measurements at different depths in the water column. Since our regression models estimate bbp at different depths, it is logical to consider that nearby values in the water column are related to each other.

Random Forest Regressor (Breiman, 2001) has been widely applied in geosciences and marine environmental studies for classification and regression tasks (Cutler et al., 2007; Ruescas et al., 2018). Regression trees are at the model's core, which effectively handles complex data when there are non-linear dependencies between a numerical response variable and a diverse set of predictors, whether qualitative or quantitative (D'Ambrosio et al., 2017). RFR is an ensemble method that combines many weak decision tree learners, which are grown in parallel to reduce the bias and variance of the model simultaneously, enhancing the model's predictive performance. Furthermore, RFR provides insights into the importance of the training features, which reveals the variables that have the most significant impact on the predictions. This capability makes the model's mechanisms and results easier to interpret and explain.

Different algorithms have been tested in previous works (see Sauzède et al. (2016, 2020)) to estimate bbp at various depths. Both works are based on a multivariate model applied to all possible outputs. In SOCA16, a Multi-Layer Perceptron is developed, while in SOCA2020 a comparison between a linear model (Ridge) and an ensemble model (Random Forest) is done. The latter showed higher performance. The Multivariate Random Forest used in this study offers higher accuracy than the univariate Random Forest, especially when the outputs are highly correlated (Schmid et al., 2022) and when complex interactions demand structured inference to be effectively managed d (Xu et al., 2019). All the previously mentioned algorithms, including Linear Regressor (LR), Ridge Linear Regressor (RLR), Random Forest Regressor (RFR), and Multi-Layer Perceptron (MLP), were tested at both 50 and 250 m depth during the dataset preparation phase. Results for 250 m are shown in Figure 2. Based on these results, the Random Forest Regressor (RFR) was selected as the most suitable algorithm for this multi-input/multi-output problem."

[Figure]

Figure 2. Comparison of different multi-output regression models for estimating vertical profiles of bbp up to 250 meters depth. Left: Depth-resolved R2 values for four regression models: Random Forest, Multi-Layer Perceptron, Ridge Regressor, and Linear Regressor. Right: Violin plots of the Mean Squared Error (MSE, log10-transformed) distributions for each model

8. In the initial paragraphs of Section 3, "Performance of the Random Forest Regressor," the authors refer to the content of Table 1, including the specific datasets corresponding to each abbreviation. However, this information should have been presented in the data introduction section. Instead, this section should provide details on the data volume obtained after feature engineering and data filtering, specifically how much data is used for training and how much for the independent validation set. This will give readers a clearer understanding of the data used in the study and its distribution between training and validation.

   We have moved this information to the data introduction section in the new version as suggested.

9. In the section "3 Performance of the Random Forest Regressor," the authors discuss the differential contribution of various features within the model. It would be beneficial to clarify the source of this feature importance data. Is it derived from the inherent parameters of the random forest model, or does it rely on additional algorithms? While the random forest, as an ensemble learning method, can assess feature importance through multiple decision trees, providing a measure of each feature's contribution to the predictive outcome, employing SHAP (SHapley Additive exPlanations) values could offer a more detailed and accurate attribution of feature importance. SHAP values provide a robust approach to explaining machine learning model outputs by assigning each feature an importance value for a particular prediction. Incorporating SHAP could enhance the transparency and depth of the analysis regarding each feature's influence on the model's performance.

   The feature importance (FI) for each model was obtained from the built-in feature importance algorithm. In this section we show the built-in FI of the different models in order to compare if the same variables are selected from the two satellite-derived products for the same training dataset (match-ups). A comparative of different feature importance algorithms, such as permutation importance and SHAP could be done, but we think that it is out of the scope of this manuscript. In the manuscript, we have clarified that it is calculated using the in-built feature importance of the random forest model.

10. In the same section, the authors depict the contribution of various features within the model. However, there are concerns regarding the clarity and utility of the presented feature importance data. Specifically, it should be clarified whether features with low contribution are consistently negligible across all depths. If these features do not significantly contribute to the model's performance at any depth, it might be beneficial to consider their removal to further reduce dimensionality and enhance the model's efficiency.

    We understand the concern regarding feature importance and dimensionality reduction. However, we have chosen to retain all features in our analysis to ensure consistency and comparability between both areas. The random forest model with fewer features is simpler, but usually similar results are yielded. Additionally, keeping all features allows for a more standardized evaluation and interpretation of results. We appreciate your suggestion and will consider discussing this point further in the manuscript to clarify our approach.

11. Additionally, some features are derived from PCA processing, and with the multitude of features used, it is challenging to distinguish between those

originating from different datasets or subjected to various treatments in the bar chart. To enhance the richness and readability of the visual information, it is suggested that the authors use distinct colors to represent bars corresponding to different types of features. This would allow for a clearer distinction between features from different datasets or processing methods, thereby providing a more informative and accessible visualization of the data. It is also worth noting that while random forest models can provide feature importances based on the model's internal assessment, these may not always reflect the true importance of features. The authors might also consider using alternative methods such as SHAP (SHapley Additive exPlanations) to calculate feature importances, which could offer a more nuanced understanding of each feature's contribution to the model's predictions.

Thank you for the suggestion. We have changed the figures using distinct colors for the different types of features in the new version.

[Figure]

*Figure 4. Feature importance for the models with S3OLCIBGC and GCGOBGC data for 50 m depth in the North Atlantic (NA) and in Subtropical Gyres (STG). Features are grouped by category according to color: (1) blue tones represent spatial and temporal descriptors, including day of year (doy), latitude, and longitude; (2) dark blue represents sea level anomaly (SLA); (3) purple indicates Mixed Layer Depth (MLD); (4) green corresponds to satellite reflectance bands from either Sentinel-3 OLCI or GlobColour with their central wavelengths; and (5) pink, red, orange, and light purple correspond to the first five principal components (PCs) derived from BGC-Argo profiles of density, temperature, salinity, and spiciness, respectively.*

12. In the concluding part of the introduction, the authors outline the main content of the research, focusing on a detailed analysis of estimating bbp in the upper layers of the ocean surface using Sentinel-3 Ocean and Land Colour Instrument (S3OLCI) data. The study aims to enhance spatial resolution from the 4 km resolution of GlobColour level-3 merged products to the 300 m Full Resolution

(FR) of Sentinel-3 OLCI. Additionally, the research evaluates model performance after incorporating OLCI spectral wavelengths as features for bbp estimation and compares these results with those obtained using GlobColour. The study also explores whether the inclusion of Inherent Optical Properties (IOPs) derived from satellite data can improve the accuracy of bbp estimation compared to using reflectances alone. These IOPs, provided by the Sentinel-3 OLCI processor, are hypothesized to significantly enhance regression models. The comparison is made between BGC-Argo data and various satellite datasets for two depth layers: from the surface to either 50 m or 250 m. However, the abstract does not provide a comprehensive and concise summary of the work and its innovative aspects. After reading the abstract, it remains unclear what the specific contributions and novelties of this research are. I recommend that the authors revise the abstract to include a brief but complete overview of the study's objectives, methods, and key findings. The abstract should clearly communicate the innovative aspects of the research, such as the use of higher resolution data, the incorporation of IOPs, and the comparison of model performances, to give readers a clear understanding of the study's significance and contributions to the field.

We have changed the abstract to highlight the main contributions and innovations of this study.

*"Abstract. As the second largest carbon reservoir on Earth, the ocean regulates the carbon balance through dissolved and particulate organic carbon forms. Monitoring carbon cycle processes is key to understanding climate system science. While most organic carbon in the ocean is dissolved, Particulate Organic Carbon (POC) plays a crucial role despite its smaller proportion, as it links surface biomass production, the deep ocean, and sedimentation. POC estimation is achieved by measuring proxies like the Particulate Backscattering Coefficient (bbp), obtained from satellite observations and in situ sensors, such as the BioGeoChemical-Argo (BGC-Argo) floats. Previous research has combined data from BGC-Argo floats and satellite sensors, demonstrating the potential of machine learning models to infer vertical bio-optical properties in the water column. By bridging the gap between surface optical properties and deep ocean processes, this approach enhances the estimation within the top 250 meters of the water column. This study focuses on such estimations with the inclusion of remote sensing data from the Sentinel-3 Ocean and Land Colour Instrument (OLCI) sensor at full resolution (300 m). The addition of optical information about absorption and scattering processes has improved the accuracy of the multi-output Random Forest models, which show promising results, especially within the first 50 meters in the Subtropical Gyres. However, in dynamic regions such as the North Atlantic, the results are less consistent, suggesting that further research is needed to understand how the complexity of the physical state of the water column modifies the bbp vertical fluxes."*

End of the introduction section:

*"Building on these results, this research proposes a more detailed analysis of estimating bbp in the upper layers of the ocean surface using the Sentinel-3 Ocean and Land Colour Instrument (S3OLCI). We enhance the spatial resolution from the 4 km resolution of GlobColour level-3 merged products (1/24° at the equator), used in previous studies, to the 300 m Full Resolution (FR) of Sentinel-3 OLCI. Additionally, we evaluate the model performance after incorporating OLCI spectral wavelengths as features for bbp estimation and*

*compare these results with those obtained using GlobColour. Another key aspect of this study is determining whether adding IOPs derived from satellite data (absorption and scattering) improves the accuracy of the bbp estimations compared to using reflectances alone. These IOPs, calculated from the OLCI processor, could significantly enhance regression models. Furthermore, bbp at different depths of the water column is estimated using multi-output models. These multi-output random forest models account for the high correlation between measurements at nearby depths. The comparison is conducted between BGC-Argo data and the satellite datasets for two depth layers: from the surface to either 50 m and 250 m.*"

13. The section "2.5 Multi-output Machine Learning Models" in the methods part of the paper should be clarified to determine whether it represents one of the study's innovative aspects. If this section indeed constitutes an innovation, it is essential to highlight it appropriately throughout the paper to ensure that readers recognize its significance. In the abstract, include a brief mention of the multi-output machine learning approach and its novelty to pique the interest of potential readers and set the stage for the detailed methodology presented later. In the introduction, provide a clear and concise explanation of what multi-output machine learning models are and how they are applied in this study. Emphasize the innovative nature of using these models, perhaps by comparing them to traditional single-output models or by discussing the advantages they offer in the research context. During the discussion, reflect on the implications of using multi-output machine learning models, including a comparison of their performance with other models, the benefits they provide in terms of accuracy or efficiency, and their potential applications in similar research endeavors. To ensure consistency and clarity, make sure that the term "multi-output" is consistently defined and used throughout the paper, and that its implications for the research are clearly articulated. If the multi-output approach is a key innovation, it should be a central theme in the narrative of the paper, guiding the reader through the methodology, results, and implications of the study.

Multi-output models are commonly used in the machine learning field as they provide a better estimation of the output variables when they are related. In this case, we are estimating the bpp at different depths of the water column. Since measurements of bpp at nearby depths are highly correlated, using a multi-output model allows us to account for this correlation effectively. We have clarified and emphasized this in the new version of the manuscript. Thank you for the comment.

Overall, addressing these suggestions will significantly enhance the manuscript's clarity, coherence, and professionalism, thereby strengthening its contribution to the field of ocean physical remote sensing.

**Citation**: https://doi.org/10.5194/egusphere-2024-3942-RC1

Finally, we would like to thank the reviewer for their valuable comments and the time and effort dedicated to reviewing our work. Your valuable suggestions have contributed to enhancing the quality and clarity of our manuscript. We sincerely appreciate your effort and expertise.

---

## Author Comment (AC2)

Dear Emmanuel Boss,

We would like to thank you for your valuable feedback and thoughtful comments on the manuscript. Your suggestions have significantly contributed to improving the clarity and quality of the work, and we sincerely appreciate the time and effort you dedicated to reviewing it.

Below, we provide responses to the reviewer's comments. The reviewer's comments are shown in black, and our responses are in blue. Paragraphs from the manuscript that have been either unchanged or modified in response to the reviewer's comments are indicated in italics. The manuscript line numbers corresponding to each comment are included in square brackets for easier reference. This new version will be uploaded into the system when permitted. Some excerpts from the text have been copied into the responses for clarity.

**[Old version Line #00]**

**1) Title and Abstract**

**[Title]** If you are interested in predicting budgets, how about the integrated POC through the euphotic depth? through the MLD?

→ We are not focused on integrating POC. The main objective of this manuscript is to estimate *bpp* using the information of the upper layers of the ocean surface from I deleSentinel-3 Ocean and Land Colour Instrument (S3OLCI) data at an enhanced full 300 m resolution. Previous studies using satellite data were conducted with GlobColour at a 4 km resolution. We use the Sentinel-3 OLCI and the GlobColour satellite products for comparison purposes.

1) **[Abstract, Line #2]** Most organic carbon in the ocean is dissolved, not particulate.

→ We have revised this sentence in the updated version of the manuscript.
**Old:** *As the second largest carbon reservoir on Earth, the ocean regulates carbon balance through Particulate Organic Carbon (POC), which links surface biomass production, the deep ocean, and sedimentation.*
**Changed to:** *While most organic carbon in the ocean is dissolved, Particulate Organic Carbon (POC) plays a crucial role -despite its smaller proportion- as it links surface biomass production with the deep ocean and sedimentation processes.*

2) **[Abstract, Line #3]** Not in the anthropogenic age. Over thousands of years, yes.

→ We have removed this sentence in the updated version of the manuscript.

3) **[Abstract, Line #4]** You write a lot about POC in the abstract but never actually compute POC in this paper.

→ Thank you for your comment. You are correct that we do not directly compute POC in this paper. Instead, we use *bpp* as a proxy for POC, which is explicitly mentioned throughout the abstract and the manuscript. We reinforce this objective in the introduction and abstract new versions. We would also like to clarify that the focus of the paper is not on POC itself but on presenting a methodology with broader applications. Mentions of POC are intended to show potential uses of the method.

**2) Introduction**

4) **[Line #21]** You find them where measured nutrients are below detection, and they divide daily (based on molecular clocks). Loss processes are as important as growth inducing ones in determining the phytoplankton, as nearly everyday the daily production is being consumed.
→ Thank you for your comment. We have revised this sentence in the updated version of the manuscript:
"*These organisms require carbon compounds, along with light and nutrients, to survive and reproduce (Fallkowski98[1], GlobalAssessment_2014). Their presence and abundance reflect the interplay of resources and losses in the environment (Behrenfeld et al., 200), with populations maintaining daily division cycles even in regions where nutrients appear to be depleted beyond detection limits (Ribalet et al., 2015; Vaulot and Marie, 1999). The observed populations represent a balance where new biomass produced each day is matched by consumption through grazing and other loss processes  (Landry and Hassett, 1982; Calbet and Landry, 2004), maintaining relatively stable populations despite continuous growth and turnover*"

5) **[Line #26]** Most of the carbon in the ocean is in dissolved organic and inorganic forms of carbon, not particulate.
→We have removed this sentence in the updated version of the manuscript.

6) **[Line #28]** Not correct. In the current state of emission of $CO_2$, one can model the ocean uptake w/o taking biology into account, simply from chemical/physical equilibration (e.g. works by Gaelan McKinley). Phytoplankton will be important once the physical/chemical equilibrium is reached, e.g. once we stopped increasing atmospheric $CO_2$. Not the current situation, unfortunately
→ We have removed this sentence in the updated version of the manuscript.

7) **[Line #29]** Reliable depends on the application, e.g. what is the tolerance for uncertainties.
→ **We have rewritten this sentence:**
"*Bio-optical sensors installed on autonomous platforms have become a **valuable** technology for acquiring in situ data on the ecological and physical status of water masses.*"
→ **We have included information about uncertainty in the bbp measurements:**
"*However, float-based measurements of bbp have a known uncertainty on the order of 10–15% (Bisson,2015). These uncertainties stem from instrumental drift, sensor calibration limitations, and the reliance on manufacturer calibration files rather than sensor-specific calibrations using dark counts. While autonomous platforms provide extensive spatial and temporal coverage, these factors must be considered when interpreting bio-optical datasets to ensure accuracy and reliability.*"
* * *
[1] Throughout this document, bibliographic references are written in the LateX format. It will be corrected in the final version.

8) **[Line #32]** Again, what are the uncertainties involved? Does it work in coastal ocean and the deep ocean where rivers and sediment resuspension of inorganic particles contribute significantly to bbp? Or when PIC is significant?

→ This question is answered in item #7. There are no coastal or deep ocean measurements in the experiments. However, we acknowledge that significant PIC concentrations can occur in regions such as the North Atlantic.

9) **[Line #33]** They depend on size, shape, internal structure and composition

→ We are referring to the water composition, not the phytoplankton

10) **[Line #35]** You are missing inelastic scattering.

→ We have rewritten this sentence in the updated version of the manuscript.

**Old:** These properties include absorption, scattering, and attenuation processes, which describe how light behaves and propagates through water

**Changed to:** "*These properties include absorption, elastic scattering, inelastic processes (such as fluorescence and Raman scattering), and attenuation, which describe how light behaves and propagates through water.*"

11) **[Line #37]** To a large degree, AOPs are only weakly dependent on light geometry.

→ The reference was taken from the Ocean Optics Web Book:

"*Apparent optical properties are those properties that (1) depend both on the medium (the IOPs) and on the geometric (directional) structure of the radiance distribution, and that (2) display enough regular features and stability to be useful descriptors of a water body.*"

→ In the new version, we have removed the sentence.

12) **[Line #62]** The uncertainty is limited to the data available. ML does a poor job extrapolating to conditions not available in the training set.

→ Like any regression method, their performance depends on the coverage of the domain and quality of the training data. Uncertainties coming from the data and model itself are expected in fitting models, but the average uncertainty can be calculated using standard error model evaluation metrics, which are applied on the training and test data. We have changed this sentence in the manuscript for better clarification.

"*This approach includes additional Sea Level Anomaly (SLA) with information about sub-mesoscale processes; it replaces satellite-derived products (bbp and chl-a) by simple reflectances at several wavelengths and explores machine learning-based techniques that are efficient at estimating retrievals, in addition to quantifying the uncertainty associated with the outputs, within the range of data on which the models have been trained. A significant improvement in the bbp predictions was revealed, especially near the surface layers.*"

13) **[Line #64]** How do you deal with the fact that bbp form floats and satellite often have significant deviations from each other (e.g. Bisson's paper)?

→ The methodology does not rely on satellite bbp retrievals as inputs. Instead, the model implicitly accounts for any discrepancies by learning the optimal function that

relates float-based bbp to the available satellite reflectances and derived IOP products.

14) **[Line #70]** These IOPs are inverted from reflectance. They do not have more information than what is available in reflectance. Here you suggest that they are somehow independent. As far as I know they are not.

→ IOPS from C2RCC are a Neural Network product. The IOPs used in the model were partly measured in situ (SIOPs), partly simulated with a bio-optical model (Hydrolight, Mobley 1994). Inelastic scattering was not included, but chl fluorescence is considered. First IOPS are inverted from the estimated reflectance, and later, there is a forward model to compute reflectance from IOPS and flag those reflectance that are out-of-scope. Our approach with the NN is that, although this derived data is not fully independent, the ability of the NN's to model non-linear relationships and the training in a big dataset of simulations may provide (or not) an improvement if compared with analytical or semi-analytical inversion models. Moreover, although the information is embedded in reflectance, derived features can help the model to handle the information efficiently, avoiding colinearities and other inherent characteristics of reflectance.

15) **[Line #71]** Not clear: do you mean the integrated POC per m^2 for the top 50 and top 250m of the ocean?
→ We do not integrate POC. We estimate the bbp at different depths.

**2.1) Study Area**

16) **[Line #80]** There is relatively little BGC-Argo data in the subtropical gyres, which makes training an ML for these regions relatively hard.
→ We are aware of that and that our dataset is not up-to-date. However, the dataset we used allows us to compare with previous studies.

17) **[Line #80]** What are 'differentiated trophic states'? How different is the vertically integrated POC between these environments?
→ We have modified the phrase**:**
"*These two areas exhibit distinct seasonal patterns throughout the year, experiencing significant differences in terms of nutrients, light availability, minimum and maximum temperature regimes, mixed layer depth (MLD) variations, thermocline levels, and mesoscale dynamics*"

18) **[Line #85]** Note that bbp has been found to also be a useful proxy of phytoplankton biomass in the upper ocean, one that does not suffer from photo-acclimation (e.g. works by Behrenfeld, Graff and Martinez-Vicente, Quu). Ignoring these work is not a good strategy for a study that wants to be taken seriously.
→ We acknowledge that our initial version did not sufficiently emphasise the relationship between bbp and phytoplankton biomass. To address this, we have added a dedicated paragraph in the **Introduction**, explicitly discussing the advantages of using bbp as a proxy for phytoplankton biomass compared to chl-a.

This revision ensures that the connection between bbp and phytoplankton is established in the introduction, but we still keep the 2.1 section focused solely on the study area description:

*"Quantifying phytoplankton biomass and carbon content is crucial to understanding these ecosystem dynamics and their role in carbon cycling. Traditionally, chlorophyll-a (chl-a) concentration has been used as a proxy for phytoplankton biomass, but its interpretation is complicated by physiological photoacclimation, which affects intracellular pigment content without necessarily indicating changes in biomass. The Particulate Backscattering Coefficient (bbp) has been recognized as a stable optical proxy for phytoplankton biomass and carbon content (Martinez-Vicente_2013, Behrenfeld_2006, Graff_2015), as it is sensitive to the abundance, size distribution, and composition of suspended particles rather than pigment concentration alone. Unlike chl-a, which can underestimate biomass in stratified and oligotrophic waters, bbp remains relatively unaffected by photoacclimation effects, making it particularly useful for studying carbon fluxes across different oceanic regions and depth layers."*

19) **[Line #86]** There is sustained surface production in all regions of the ocean (except extreme high latitudes during polar night), even when nutrient levels are undetectable.
→ The idea here is to remark on the differences between those two study areas, not the similarities. We can say something about it, but also about species or sizes and how important this is for a balanced food chain and conservation of ecosystems.

20) **[Line #88]** Yet molecular clocks show that the cells there divide daily. How come? And when you integrate vertically the biomass, the values are not orders of magnitude less than in the productive NA.

→We do not have integrated bbp in this experiment.

**2.2) BGC-Argo Data**

21) **[Line #103]** Most floats have significantly poorer vertical resolution.
"The BGC-Argo floats usually collect measurements from 1,000 m to the surface, with a depth resolution of ~1 meter, every 10 days."

We utilized a dataset prepared by Dr. Sauzéde, which has been featured in previous publications.This dataset includes floats from amol, bodc, coriolis, csiro and incois, not only French floats. The bbp profiles have been linearly interpolated onto the fixed output depths of retrieval, making the resolution homogeneous across all floats. We have also included this description in the manuscript.

22) **[Line #104]** This definition of the euphotic depth is useless. Phytoplankton care about absolute photon flux, not the relative one (e.g. Sverdrup, 1943, Behrenfeld and Boss, 2017). Recent studies using BGC-Argo suggest phytoplankton can grow at

extremely        low        light        levels        of        light,        e.g.
https://www.science.org/doi/10.1126/sciadv.abc2678
→ We considered following an approach that focuses on the optical properties of the water column as a physical phenomenon rather than on biological responses to light availability, although we derive bbp. We think a light-field-based definition provides a consistent boundary for analysing water column dynamics in the two regions and allows standardised comparisons across our datasets. However, we have included references addressing phytoplankton growth under varying or low light conditions to acknowledge the biological importance of absolute light levels.

23) **[Line #107]**  The flux of sinking carbon…Not in this day an age. Read papers by biogeochemists such as McKinley and Gruber.
→ We have removed this sentence.

24) **[Line #108]** Describe to the readers why spiciness is important and what it is. Most do not know it.
→ Thank you for pointing this out. We have added a description of spiciness in the revised version **:**.
*"Spiciness reflects density-compensated variations in temperature and salinity, providing a tracer for water mass origins and mixing processes (Smith_2009). Since particle concentrations and optical properties often differ between water masses, spiciness anomalies can be associated with variations in the bbp. Warmer and saltier waters (higher spiciness) can enhance stratification, reducing vertical nutrient fluxes and potentially limiting biological production, leading to lower concentrations of organic particulate matter and thus lower bbp."*

**2.3) BGC-Argo and Satellite Match-up Databases**

25) **[Line #111]** Not from all the BGC-floats. Are you only  using the French ones?
→ We have floats from aoml, bodc, coriolis, csiro and in, not only French floats. See #21.

26) **[Line #119]** Your keeping 3 points after the decimal suggest your uncertainty is on the order of 0.1%. In fact, it is likely larger than 10% (bbp estimation)
→ The methodology was taken from the SOCA2020 method, and the processing is explained in Sauzède et al. (2016, 2020). These values come from it.

27) **[Line #126]** Are you using all available BGC-Argo float with bbp or only the French ones which have more vertical resolution? If you are using all of them, how do you change resolution?
→ We used a dataset that was prepared by Dr. Sauzéde and used in previous publications. This dataset includes floats from amol, bodc, coriolis, csiro and incois, not only French floats. See #21.

28) **[Line #133]** How will it change if you used more stringent criteria, such as those of Bailey and Werdell (5km, 3hrs)? BBp is observed to have diel cycles in many places in the ocean, and using all measurements may introduce bias.

→ Initially, our database also considered a 5-day window, and reducing the time constraint to 24 hours decreased the dataset from 4,115 to 763 samples. Given this significant reduction, further tightening the time criteria would compromise the dataset even further, making it challenging to maintain a sufficient number of samples for model training.

29) **[Line #138]** How far do you expect a water parcel in the ocean will move in 24hrs (typical ocean currents are ~10cm/s).
→ It will depend on the position of the float, on the horizontal currents in the area, on the wind intensity, and on latitude - to take into account other effects (e.g Ekmann spiral, geostrophic currents, etc.).

30) **[Line #139]** Once the match-up between satellite and float is performed…what does it mean?
→Modified to:
 *"Once the match-up between satellite and float is established, a baseline quality control is applied to ensure that the satellite-measured reflectances are not affected by sensor noise or transient atmospheric disturbances, maintaining radiometric consistency"*

31) **[Table 1]** Why use only PAR at the surface? What about the euphotic depth (as determine from an isolume)? Wouldn't you think subsurface light matters?
→ PAR comes from satellite observations (GlobColour), not from floats (Table 1). The main idea behind SOCA is to only use Argo (T/S) floats. Because SOCA does not integrate float parameters as inputs because the application is independent from BGC-Argo floats and only uses satellite + Argo (T/S data). If we use the euphotic depth determined from the float, it severely restricts the applications.

32) **[Line #141]** Are you log transforming some of the measurements?
→Yes, we apply a base-10 logarithmic transformation to the bbp to reduce data skewness and improve algorithm performance. All metrics and plots are subsequently transformed back to the original linear scale from log space. It is properly described in the data section.

33) **[Line #145]** This is very little data. Are you sure you used all that is available in those waters? You could have also used the Med where there is the most data and coverage.
→ We are using data of opportunity. Using the same dataset as in another published work allows us to compare results properly. We could plan to add more data from other regions, and we would be happy to do it if something is already available in a new project.

34) **[Line #156]** Are the IOPs derived from the reflectance data? If the answer is yes, how can they provide independent information the the ML?
→ Explained in item #14

**3.1.1) Shallow waters: from 0 to 50 meters depth**

**35) [Line #224]** R^2 a good metric? It is typically very influenced by the dynamic range of the variables involved and provide no information on goodness of fit.

→ R2 is not a good metric by itself, but we also calculated MAE (Mean Absolute Error) and MAPD (Median Absolute Percentage Deviation). R2 is also used profusely within both the OC and the ML communities. To complete the overview, we have added the bias and the relative error on the profiles and the Median Absolute Percentage Deviation (MAPD) in statistics in Table 3-4, Figure 3 (50m depth, S3OLCIBGC and GCGOBGC), Figure 5 (250m depth, S3OLCIBGC and GCGOBGC) and Figure 7 (250m depth, S3OLCI and S3IOPS).

*Table 3. Statistics by region at 50 m and 250 m depth models with satellite and BGC-Argo. Median Absolute Percentage Deviation (MAPD) is expressed in % and Mean Absolute Error (MAE) in $m^{-1}$.*

| Depth | Region | | GCGOBGC | S3OLCIBGC | S3IOPs | S3OLCI |
|---|---|---|---|---|---|---|
| **50 m** | North Atlantic | $R^2$ | 0.72 | 0.78 | 0.74 | 0.77 |
| | | MAPD | 8.19 | 10.77 | 13.46 | 12.96 |
| | | MAE $(\times 10^{-4})$ | 3.11 | 2.86 | 3.04 | 2.89 |
| | Subtropical Gyres | $R^2$ | 0.87 | 0.86 | 0.88 | 0.84 |
| | | MAPD | 5.60 | 5.54 | 5.61 | 5.56 |
| | | MAE $(\times 10^{-5})$ | 4.16 | 4.50 | 4.39 | 4.81 |
| **250 m** | North Atlantic | $R^2$ | 0.84 | 0.81 | 0.80 | 0.80 |
| | | MAPD | 3.37 | 5.24 | 6.38 | 6.18 |
| | | MAE $(\times 10^{-4})$ | 0.85 | 1.02 | 1.12 | 1.09 |
| | Subtropical Gyres | $R^2$ | 0.90 | 0.89 | 0.88 | 0.88 |
| | | MAPD | 4.97 | 5.36 | 5.98 | 5.47 |
| | | MAE $(\times 10^{-5})$ | 3.19 | 3.46 | 3.74 | 3.74 |

*Table 4. Validation with independent floats by region at 50 m and 250 m depth models with satellite and BGC-Argo data. MAPD is expressed in %, MSE and RMSE in $m^{-1}$.*

| Depth | Region | | GCGOBGC | S3OLCIBGC | S3IOPs | S3OLCI |
|---|---|---|---|---|---|---|
| **50 m** | North Atlantic | $R^2$ | 0.26 | 0.44 | 0.44 | 0.41 |
| | | MAPD | 38.48 | 33.74 | 27.96 | 32.5 |
| | | MAE $\times 10^{-4}$ | 7.00 | 5.91 | 5.61 | 5.67 |
| | | RMSE $\times 10^{-4}$ | 9.05 | 7.38 | 7.05 | 7.13 |
| | Subtropical Gyres | $R^2$ | 0.65 | 0.63 | 0.64 | 0.63 |
| | | MAPD | 5.06 | 4.90 | 5.69 | 5.99 |
| | | MAE $\times 10^{-5}$ | 2.93 | 3.08 | 3.09 | 3.12 |
| | | RMSE $\times 10^{-5}$ | 3.60 | 3.73 | 3.67 | 3.73 |
| **250 m** | North Atlantic | $R^2$ | 0.32 | 0.31 | 0.29 | 0.29 |
| | | MAPD | 6.56 | 6.38 | 6.41 | 6.37 |
| | | MAE $\times 10^{-4}$ | 7.50 | 7.68 | 7.44 | 7.47 |
| | | RMSE $\times 10^{-4}$ | 10.4 | 10.5 | 10.6 | 10.7 |
| | Subtropical Gyres | $R^2$ | 0.58 | 0.53 | 0.54 | 0.56 |
| | | MAPD | 7.38 | 8.34 | 8.28 | 8.06 |
| | | MAE $\times 10^{-5}$ | 3.88 | 4.19 | 4.18 | 4.02 |
| | | RMSE $\times 10^{-5}$ | 4.92 | 5.32 | 5.34 | 5.13 |

[Figure]

**Figure 3.** Model performance for estimating shallow water $b_{bp}$ profiles (0–50 m). **(A)** Depth-resolved metrics comparing model predictions using S3OLCIBGC and GCGOBGC as inputs: Coefficient of determination ($R^2$), Mean Absolute Error (MAE), and bias. Shaded horizontal lines indicate the average Mixed Layer Depth (MLD) per region. **(B -C)** Measured and predicted $b_{bp}$ profiles in NA **(B)** and STG **(C)** using S3OLCIBGC. The rightmost bars show the mean relative error by depth.

[Figure]

**Figure 5.** Model performance for estimating deep water $b_{bp}$ profiles (0–250 m). **(A)** Depth-resolved metrics comparing model predictions using S3OLCIBGC and GCGOBGC as inputs: Coefficient of determination ($R^2$), Mean Absolute Error (MAE), and bias. Shaded horizontal lines indicate the average Mixed Layer Depth (MLD) per region. **(B -C)** Measured and predicted $b_{bp}$ profiles in NA (B) and STG (C) using S3OLCIBGC. The rightmost bars show the mean relative error by depth.

[Figure]

**Figure 7.** Model performance for estimating shallow water $b_{bp}$ profiles (0–50 m) with only satellite data. **(A)** Depth-resolved metrics comparing model predictions using S3OLCI and S3IOPS as inputs: Coefficient of determination ($R^2$), Mean Absolute Error (MAE), and bias. Shaded horizontal lines indicate the average Mixed Layer Depth (MLD) per region. **(B-C)** Measured and predicted $b_{bp}$ profiles in NA (B) and STG (C) using S3OLCI. The rightmost bars show the mean relative error by depth.

**36) [Line #224]** Are three points after the decimal significant?

→ The interesting thing is to look at the values at different depth layers in Figure 2 and see how the models behave differently at different depths due to the different inputs, although the general metrics are pretty similar.

**37) [Line #233]** What do you mean by 'good quality'. Can you quantify it?

→ We are using the statistics derived from the test/validation, and based on that (coefficients and errors), we can say that the information provided adds value to the results. We have replaced "good quality" with the word "meaningful". We hope this in fine.

**38) [Line #237]** This is the value at the surface. Since you are deriving a vertical profile it may have limiting information w/o more about the vertical attenuation of the light.

→ Sure. The research aims to determine if ML algorithms can use surface optical information, together with information about non-optical parameters, such as salinity, density, temperature, and spiciness, to derive in-depth optical properties up to a specific depth. Also, to evaluate the algorithm level of importance of the surface reflectance for estimating bbp at the two different depths.

**39) [Line #241]** Here you are speculating. These assertion can be tested, e.g. by providing seasonality. DOY with latitude is directly linked to daily PAR and solar zenith angle.

→We think we are not speculating, as the seasonality of phytoplankton in NA is well known. We have included some bibliographic references to support this hypothesis about why this feature is relevant for the model **[Line 263 - 265]**:

*"The feature DOY, which accounts for the temporal component, reflects the seasonality that affects phytoplankton cycles and, consequently, the POC dynamics in these regions. This seasonality has been studied in various works, such as Honjo_1993 and Sanders_2014"*

**40) [Figure 2]** Can you also provide the relative errors? Units? The average reader will not know how to read your plots. There are no units anywhere (e.g. m^-1).

→Thanks for pointing it out. We have changed the violin plot to depth-resolved model bias, providing the values in bbp units. We have also included the units in the profiles and an additional plot for depth-resolved relative errors.

See response #35 for tables and figures.

**41) [Line #245]** Do you mean the maximal surface biomass? The maximal vertically integrated biomass?

Modified to:

*"The maximum mixed layer phytoplankton biomass usually occurs from June to August, coinciding with higher water temperatures (Yang2020)"*

**42) [Line #255]** Are vertical gradients in R^2 the same as vertical gradients in bbp?

→ Vertical gradients in R^2 for the NA region are correlated with the increase of the errors with depth. It can be observed in the new figures, which are included in response #35. Right bar plots show the mean relative errors with depth.

**43) [Line #263]** is high biomass a sign of a better ecology? What is the meaning of this value judgement? Based on it a trophic outflow of a river laden with chemicals from aquaculture is the best ecology.

→ Our intention was not to imply a value judgement about ecosystem quality or health. We referred to "more favourable ecological conditions for phytoplankton" specifically in the context of conditions that support higher phytoplankton growth (e.g., due to nutrient input from upwelling or water mass divergence), not as an indication of overall ecological integrity. To avoid confusion, we have rephrased this sentence for clarity:

*"At the edges of the gyres—near eastern and western boundary currents, subpolar and equatorial regions—nutrient-rich waters are often introduced by divergence and upwelling, creating conditions that are more favorable for phytoplankton growth"*

**44)** Figure 3 too small

→ We have changed it for a better visualisation, and we use distinct colours for the different types of features in the new version.

[Figure]

*Figure 4. Feature importance for the models trained with S3OLCIBGC and GCGOBGC to estimate bbp in shallow waters (0–50 m) in the NA and STG. Features are grouped by category according to colour: (1) blue represent spatial and temporal descriptors, including day of year (doy), latitude, and longitude;*

*(2) dark blue represents sea level anomaly (SLA); (3) purple indicates Mixed Layer Depth (MLD); (4) green corresponds to satellite reflectance bands from either Sentinel-3 OLCI or GlobColour with their central wavelengths; and (5) pink, red, orange, and light purple correspond to the first five principal components (PCs) derived from BGC-Argo profiles of density, temperature, salinity, and spiciness, respectively.*

**3.1.2 Deep waters: from 0 to 250 meters depth**

**45) [Line #273]** Aren't they also dynamic in the STGs?
→ We are just trying to describe the NA observations.

**46) [Line #274 - #297]**I thought the MLD is dynamic and changes throughout the year, even more so in the NA.
→MLD is dynamic, but in our observations, the mean depth for the MLD was 30 m. We have pointed out this information in the new version to avoid misunderstandings.

*"A decrease in accuracy is observed around 30 meters, coinciding with the average MLD observed in our dataset (Figure (A)). While the MLD is inherently dynamic and varies throughout the year, this mean depth represents a critical boundary in our observations"*

**47) [Line #278]** You keep mixing opinions and results. Separate your discussion from results so it is clearer to the reader what you
→We have removed our interpretation of the results in this section.

**48) [Line #281]** I thought SLA reflects horizontal changes in properties, e.g. eddies.
→This feature is not discussed in the new results, as we focus in greater detail on other, more relevant features

**49) [Line #285 ] (STG)** I thought the water column was highly stratified there, or is it that while the water is stratified bbp is more homogeneous, as was seen in Kitchen and Zaneveld 1990 from beam attenuation.
→ We have changed this sentence according to the comments:
*"In the STG region, model performance exceeds that of the North Atlantic, likely due to the more optically homogeneous conditions despite the strongly stratified nature of these oligotrophic waters (Figure 5C). Similar patterns have been observed in stratified waters, where optical properties like beam attenuation remain relatively homogeneous (Kitchen_1990).*

**50) [Line #287]** In many regions the relative increase in actual biomass is minimal, see Kitchen and Zaneveld 1990. The relative change with depth < 10%. What kind of relative changes do you see in bbp/POC?
→ We have changed this sentence according to the comments.

**51) [Line #291]** Look at integrated bbp. You be surprised to find they are much less different than you think based on surface bbp values.
→ This is fact good news since it supports the validity of the surface estimation of bbp when trying also to understand what is happening in depth.

**3.2 S3OLCI: results of Sentinel-3 OLCI without BGC-ARGO data**

**52) [Line #297]** Is this any surprise?

→Not really, we are just describing the results.

**53) [Line #302]** Is this significant? Again, if the IOP source is reflectance one would expect no difference…

→ IOPs are not directly derived from reflectance. See response #14 for more details.

**54) [Line #312]** Again, please separate results from discussion/speculations for easier reading.

→ We have removed our interpretation of the results.

**55) [Figure 6]** Interesting that the distribution of measured has a single peak while the predicted has two.

→ The bimodality in predicted bbp probably reflects the limited representation of transition values in the training data.

**56) [Figure 6]** Will be useful to have RMSE stats as well as MSE.

→ We have added new statistics to complete the overview. New tables and figures are in response #35.

**57) [Line #332]** Is it true in a relative sense as well? Also, based on Fig. 6 the histogram of distributions seems better for NA while that for predicted STG is bi-modal a feature not seen in the measured data.

→ We have included MAPD (Median Absolute Percentage Deviation) to have relative references. See response in item #35.

**58) [Table 4]** You have not defined MAE anywhere (I assume it is the mean absolute error but I may be wrong as it has no units here).

→ Thanks for pointing it out. It is defined now, together with other statistics used.

**59) [Table 4]** Please add RMSE and keep only significant units. Provide the units to the data that has units.

→ Thank you for pointing it. It has been included together with Median Absolute Percentage Deviation (MAPD), depth-resolved bias plots, and Mean Absolute Error (MAE). See response in item #35 with the modified tables and figures as suggested.

**60) [Line #340]** Are ML model not empirical?

→ Yes, they are considered empirical if we understand them as data-driven approaches. However, we refer here to classical/traditional statistical methods such as univariate t-test, ANOVA, Pearson's and Spearman correlation, linear regression, etc. We have clarified it in the manuscript.

**61) [Line #355]** How do you explain the fact that surface populations are observed to divide at near maximal rates in those waters?

→ We would prefer not to speculate. We would be grateful if you could help us with this explanation.

**62) [Line #360]** That was in the predicted data, not the measured.

→ You are right, but it is related to the lack of samples for the same range in the in situ data that we used to train the model. Since this is an independent float, the distribution of the in situ data of this float could be not represented in the training set.

**63) [Line #365]** Does it all sink? A small portion? STG are known to be regions with very efficient recycling of nutrients.

→We have slightly modified the sentence with the aim of making it clearer:

*"This organic matter is subsequently transformed through respiration, particle aggregation, zooplankton grazing, feces production, and microbial decomposition (Siegel et al.,2014), before eventually a fraction of it sinks to deeper layers"*

**64) [Line #373]** Which database? Why didn't you use an up-to-date one?

→ We are working with "data of opportunity". The dataset was ready, and previous works have used it, so they were available for comparison. We will be happy to apply the model to extend the dataset, if available. Always open to collaborate!

**65) [Line #377]** ??? Not clear. Bisson was focused on bbp at the surface. You could say that if you compared the same deliverable, rather than the depth distribution.

→We were referring to the SOCA2020, which shows a similar approach.

**66) [Line #378]** Why do you expect a relationship between CDOM and POC?

→ Increase in photoproduction of CO2 from CDOM  (Belanger et al. 2006) due to increase in UV radiation and decrease in sea ice. CDOM refers to the yellow substance, which is a fraction of the DOC pool. Organic carbon is separated into POC and DOC. POC has a high seasonality, and half of the primary production is channeled into DOC via direct release, sloppy feeding, or after the death of phytoplankton. In the open ocean DOC exceeds POC by an order of magnitude.

POC can be estimated over the global ocean with satellite data (Liosel et al. 2002, Stramski et al. 2008); however, DOC estimations are still challenging (Aurin et al., 2018 and Siegel et al., 2002). There is a potential use of aCDOM to improve DOC estimations (especially in coastal waters)  together with physical variables like SST or salinity. If CDOM can really improve DOC estimations and we can do it globally with satellites, a better understanding of the relationship between DOC and POC can also be analysed temporally and spatially.

**67) [Line #380]** Where? Did you use all the floats profile available? If yes, for what dates? Best if you provide a data repository in Zenodo so other can replicate your results.

To better compare our results with the SOCA2020 method presented in Sauzède et al. (2020), we used the same database, specifically including all available bbp profiles up to 2018 from two restricted areas that represent two distinct trophic regimes: the North Atlantic Ocean (NA) and the oligotrophic Subtropical Gyres (STG).

**68)** **[Line #381]** If you want others to use your result (and cite you), provide the ML model you have created.

→ We will provide the code and model using GitHub on the ISP site in accordance with our group policy of publishing developed models in open access (https://github.com/IPL-UV).